# RIDE: Difficulty Evolving Perturbation with Item Response Theory for Mathematical Reasoning

## Abstract

Large language models (LLMs) achieve high performance on mathematical reasoning, but these results can be inflated by training data leakage or superficial pattern matching rather than genuine reasoning. To this end, an adversarial perturbation–based evaluation is needed to measure true mathematical reasoning ability. Current rule-based perturbation methods often generate ill-posed questions and impede the systematic evaluation of question difficulty and the evolution of benchmarks. To bridge this gap, we propose RIDE, a novel adversarial question-rewriting framework that leverages Item Response Theory (IRT) to rigorously measure question difficulty and to generate intrinsically more challenging, well-posed variations of mathematical problems. We employ 35 LLMs to simulate students and build a difficulty ranker from their responses. This ranker provides a reward signal during reinforcement learning and guides a question-rewriting model to reformulate existing questions across difficulty levels. Applying RIDE to competition-level mathematical benchmarks yields perturbed versions that degrade advanced LLM performance, with experiments showing an average 21.73% drop across 26 models, thereby exposing limited robustness in mathematical reasoning and confirming the validity of our evaluation approach.

## 1 Introduction

Mathematical reasoning is widely regarded as a crucial testament for evaluating the reasoning capabilities of large language models (LLMs). Recent works like DeepSeek-Math (Shao et al., 2024) and Qwen2.5-Math (Yang et al., 2024) clearly demonstrate the benefits of reinforcement learning for mathematical reasoning. Nevertheless, strong performance in mathematical reasoning may not directly reflect the model's reasoning ability (Wu et al., 2025). It may instead result from data contamination during pretraining (Golchin & Surdeanu, 2024) that leads to memorization-based answers (Zhang et al., 2024; Li et al., 2025), or from the model relying primarily on superficial pattern matching without truly understanding mathematical knowledge (Li et al., 2024; Mirzadeh et al., 2025). This phenomenon undermines the reliability of LLMs as genuine mathematical reasoners.

To address this issue, there is an urgent need for a more adversarial evaluation benchmark that can faithfully characterize the robustness of mathematical reasoning. For example, a rule-based rewriting method is proposed (Li et al., 2024; Zhou et al., 2024), which applies operations such as numerical substitution and symbol modification to reformulate questions. This method has been shown to reduce model performance. However, rule-based approaches may render the rewritten questions ill-posed and unsolvable (Xue et al., 2025), thereby undermining their validity as reliable benchmarks. Moreover, previous work has focused mainly on low-difficulty datasets Cobbe et al. (2021), with limited attempts at more challenging data. Based on this, we identify the following challenges: (1) **Robust Adversarial Perturbation.** Robust perturbations to the data need to both degrade the model performance and remain credible and reasonable, which requires shifting from injecting noise to directly increasing the difficulty of the questions as a means of perturbation. Moreover, the perturbation should be unpredictable for changing, and simple rules cannot capture the evolution. (2) **More Challenging Data.** The perturbation of data should not be limited to the same difficulty level of mathematical reasoning. It is also necessary to explore its effectiveness on difficulty evolution, ensuring reasonable perturbations can be applied to mathematical competition

questions. These challenges require us to perturb existing benchmarks in ways that substantively increase the intrinsic difficulty of the questions (Huang et al., 2025). A question's difficulty is often defined by whether a model answers correctly (Lan et al., 2024), but this definition depends heavily on the selected models, and major voting across these models ignores the individual differences in capabilities. Besides, defining difficulty by the steps of the reasoning chain (Yu et al., 2025) is easily misled by shortcut reasoning and does not account for the numerical computation complexity specific to mathematical reasoning.

To this end, we propose **RIDE**, a benchmark perturbation rewriting framework that incorporates Item Response Theory (IRT) to measure question difficulty (Vania et al., 2021; Scarlatos et al., 2025). Specifically, 35 cutting-edge LLMs are treated as students to answer $2,000$ questions sampled from the high-difficulty mathematical dataset DeepMath (He et al., 2025). A statistical model jointly characterizes the latent abilities of the student models and the difficulty parameters of the questions. Based on the IRT-estimated difficulty, a pairwise ranker is trained: given question embedding vectors, it outputs the probability that one question is more difficult than another. The ranker's output fur-

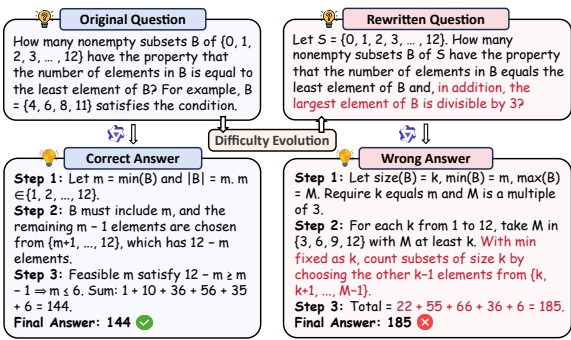

Figure 1: Answers of Qwen3-30B-A3B to a mathematical question and variation rewritten by RIDE.

ther serves as a difficulty reward signal for reinforcement learning on Qwen3-8B, resulting in the question rewriting model RIDE-8B. Applying reinforcement learning to difficulty-evolving question rewriting encourages the model to adaptively generate questions with higher difficulty, thus avoiding superficial modifications, while further enhancing consistency between the rewritten questions and their answers by setting a correctness reward. RIDE-8B is then applied to rewrite competition-level mathematical benchmarks such as AIME-24 and AMC-23, achieving difficulty evolution for these datasets. Experiments demonstrate that the perturbed benchmarks induce performance drops in frontier reasoning models, including GPT-5, DeepSeek-V3.1, and others, thereby enabling robust evaluation and comparison of advanced models' mathematical reasoning abilities (as shown in Figure 1). In addition, RIDE can be used as a data augmentation method to enhance training data and improve mathematical reasoning performance. Our contributions are as follows:

- We propose a benchmark perturbation framework with IRT to evaluate robustness in mathematical reasoning, reducing partial data leakage during pretraining on evaluation. We measure robustness via performance degradation and observe that most LLMs are sensitive to perturbations, indicating poor robustness.

- We propose a more scalable IRT-based difficulty estimation method and a corresponding difficulty ranker. In addition, we introduce a reinforcement learning scheme for training a question-rewriting model.

- We release a question rewriting model (RIDE-8B), two rewritten competition-level benchmarks (RIDE-AIME and RIDE-AMC), and a training dataset of over 10K rewritten questions (RIDE-DeepMath).

## 2 RELATED WORK

**Robustness Evaluation.** Existing LLM robustness evaluations (Ul Abedin et al., 2025) often use controlled perturbations to test whether models truly understand tasks rather than memorize surface patterns (Li et al., 2024; Yang et al., 2025a). A common practice is to construct original items and perturbed counterparts around the same semantics and compare the resulting performance drop (Goel et al., 2021; Chang et al., 2023; Hou et al., 2025; Yang et al., 2025b). For example, GSM-PLUS Li et al. (2024) applies rule-based perturbations such as numerical substitution (Shrestha et al., 2025) and operation modification, revealing substantial degradation in many LLMs. Unlike rule-based methods, we adapt a reinforcement learning-based rewriting approach

to generate perturbations, producing better-formed data, avoiding unsolvable or multiple-solution cases, and increasing unpredictability.

**Item Response Theory (IRT).** Originating from educational measurement, IRT is widely applied in AI for automated item difficulty estimation (Feng et al., 2025; Chen & Shiu, 2025) and efficient evaluation (Polo et al., 2024; Martínez-Plumed et al., 2022). Traditional IRT depends on real student responses, which are often difficult to obtain due to privacy constraints. Prior work addresses this by simulating response matrices (Lalor et al., 2019; Truong et al., 2025) or predicting difficulty from text (Martínez-Plumed et al., 2022), which leads to high computational costs (Truong et al., 2025). We mitigate these issues via data augmentation to reduce the number of required student models and reframing difficulty estimation as a pairwise ranking task to reduce numerical prediction errors.

## 3 PRELIMINARY

We briefly introduce the IRT-based difficulty in our method. Consider a test taker with fixed ability $\theta$ and a math reasoning question $q$ with a difficulty score $d$. The response $y$ is modeled as a Bernoulli random variable that indicates correctness, where $y = 1$ denotes a correct answer and $y = 0$ denotes an incorrect answer. We introduce the Rasch model (Rasch, 1993), a foundational one-parameter (1-PL) IRT model in which the probability of a correct response is the logistic function of the difference between the test taker's ability $\theta$ and the item's difficulty $d$:

$$\Pr(y = 1 \mid \theta, d) = \sigma(\theta - d) = \frac{1}{1 + e^{-(\theta - d)}}$$

Given a binary response matrix $\mathbf{Y} \in \{0, 1\}^{M \times N}$, where $M$ and $N$ denote the number of test takers and questions, respectively. Each entry $Y_{ij}$ indicates the $i$-th test taker's answer to the $j$-th question. Based on this response matrix, one can estimate the ability parameters $\theta$ and the difficulty parameters $d$ using a variety of statistical inference methods, such as the Bayesian inference via Markov chain Monte Carlo (MCMC) sampling (Chen et al., 2021) and variational inference (VI) (Blei et al., 2017). In this paper, we apply the VI method provided by *py-irt* (Lalor & Rodriguez, 2023). Concretely, we fit the Rasch model via stochastic variational inference on the response matrix with a mean-field Normal guide for $\theta$ and $d$, maximizing the evidence lower bound $\mathcal{L}$:

$$\text{KL}(g\|p) = \sum_{i=1}^{M} \text{KL}\big(g(\theta_i) \,\|\, p(\theta_i)\big) + \sum_{j=1}^{N} \text{KL}\big(g(d_j) \,\|\, p(d_j)\big)$$

$$\max_g \mathcal{L} = \sum_{i=1}^{M} \sum_{j=1}^{N} \mathbb{E}_g\big[\, Y_{ij}(\theta_i - d_j) - \log\big(1 + e^{\theta_i - d_j}\big)\big] - \text{KL}\big(g \,\|\, p\big)$$

Here, $g$ denotes the variational joint distribution over the latent parameters and $p$ denotes the prior joint distribution over $(\theta, d)$. $\text{KL}(g\|p)$ denotes the Kullback-Leibler divergence from $g$ to $p$. We take posterior means as point estimates from $g$ to obtain IRT-based difficulty $d$ for each question.

## 4 METHOD

In this section, we elaborate on the details of RIDE-Math. Inspired by the IRT-based difficulty estimation method with LLMs acting as students (Truong et al., 2025), we construct a difficulty ranker for math questions by performing parameter estimation described in Section 3. This ranker serves as a reward model by providing difficulty reward signals, which are used to guide reinforcement learning on the existing high-difficulty dataset DeepMath (He et al., 2025). Through this rewriting process, we are able to generate difficulty in evolving math reasoning data. The overall architecture of our method is shown in Figure 2

### 4.1 IRT-GUIDED DIFFICULTY RANKER

#### 4.1.1 IRT PARAMETER ESTIMATION

We draw $N = 2,000$ math questions from the DeepMath dataset to construct a quiz. We then employ $M = 35$ cutting-edge LLMs with different parameter scales and architectures as students to

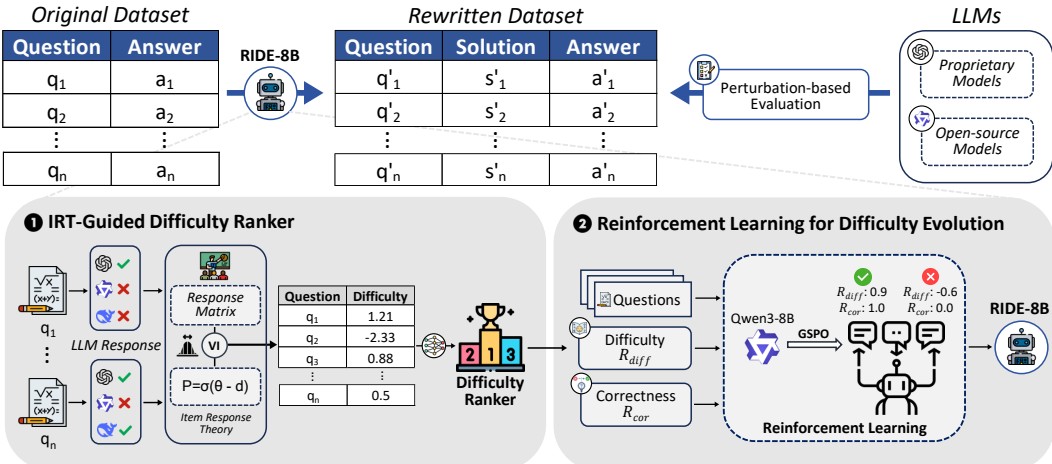

Figure 2: The overall architecture of RIDE.

solve these questions, thereby constructing a response matrix $\mathbf{Y} \in \{0,1\}^{M \times N}$. Specifically, for a math question $j$, the response vector can be described as

$$\mathbf{v}^j = [\mathbb{I}[g^j = y_1], \mathbb{I}[g^j = y_2]..., \mathbb{I}[g^j = y_M]]^\intercal$$

Here, $\mathbb{I}$ denotes the 0–1 indicator function, $g^j$ denotes the ground-truth answer to the $j$-th question, $y_i$ represents the response of the $i$-th student model. After obtaining the response vectors for all $N$ questions, we combine them to form the response matrix $Y$:

$$\mathbf{Y} = \left[ \mathbf{v}^1, \ \mathbf{v}^2, \ \dots, \ \mathbf{v}^N \right]$$

As a limited number of student models is not enough to estimate IRT parameters accurately, while employing a large number of LLMs as students is computationally costly, we augment the student–response data to simulate additional student behaviors. We adopt two augmentation strategies:

- **Variational Autoencoder (VAE) method.** We mitigate the scarcity of student responses by training a VAE (Kingma & Welling, 2022) on the observed response matrix $\mathbf{Y}$ to learn a low-dimensional manifold of student behavior. During generation, we draw a latent code $\mathbf{z}$ from the Gaussian prior and decode it into per-question correctness probabilities $\pi(\mathbf{z})$, from which Bernoulli draws yield a synthetic student-question response vector with conditional independence across items given $\mathbf{z}$:

$$\mathbf{r}_{\text{VAE},j} \mid \mathbf{z} \sim \text{Bernoulli}\big(\pi_j(\mathbf{z})\big), \qquad j = 1, \dots, N$$

- **Sampling method.** We simulate the responses of students by drawing from the empirical distribution observed in $\mathbf{Y}$. For each question $j$, we compute the empirical correctness rate $C$ over the observed responses and use it as the Bernoulli success probability:

$$\mathbf{r}_{\text{Sampling},j} \sim \text{Bernoulli}\big(C_j\big), \qquad j = 1, \dots, N$$

Each sampled $\mathbf{r}_{\text{VAE}}$ and $\mathbf{r}_{\text{Sampling}}$ is then appended as a new student row to $\mathbf{Y}$.

These methods let us synthesize student–item response vectors that respect observed item statistics and student tendencies. We concatenate the synthetic responses $\mathbf{r}_{\text{VAE}}$ and $\mathbf{r}_{\text{Sampling}}$ with the originals to form an augmented matrix $\tilde{\mathbf{Y}} \in \{0,1\}^{M+S+T,N}$:

$$\tilde{\mathbf{Y}} = \begin{bmatrix} \mathbf{Y} \\ \mathbf{R}_{\text{VAE}} \\ \mathbf{R}_{\text{Sampling}} \end{bmatrix}, \quad \mathbf{R}_{\text{VAE}} = [\mathbf{r}^1_{\text{VAE}}, ..., \mathbf{r}^S_{\text{VAE}}]^\intercal, \quad \mathbf{R}_{\text{Sampling}} = [\mathbf{r}^1_{\text{Sampling}}, ..., \mathbf{r}^T_{\text{Sampling}}]^\intercal$$

$S$ and $T$ denote the number of synthetic students by VAE and sampling, respectively. Then the modeling ability of IRT calibration for question difficulties can be improved. We perform IRT parameter estimation based on the response matrix $\tilde{\mathbf{Y}}$. Following the parameter estimation method described in Section 3, we obtain an IRT-estimated difficulty $d$ for each question and construct $N$ question-difficulty pairs $(q, d)$ for training the difficulty ranker, where $q$ denotes the question.

### 4.1.2 PAIRWISE DIFFICULTY RANKER

In traditional IRT, estimating the difficulty of a new question requires collecting fresh student response data. To endow IRT difficulty estimation with generalization to unseen questions, we adopt an amortized approach (Truong et al., 2025), which extracts content features via an embedding model and learn a mapping from question content to difficulty value. However, due to the domain-specific symbolic semantics of math questions (Mirzadeh et al., 2025), directly regressing from embeddings to a scalar difficulty may result in unreliable performance. Therefore, instead of regressing the difficulty value, we frame the task as a pairwise ranking question, training a ranker to identify the more difficult question from a pair of question embeddings. We use OpenAI's text-embedding-3-large [1] model to obtain embedding vectors for each question-difficulty pair $(q, d)$, creating $N$ question-embedding-difficulty triplets as $(q, e, d)$. We then group these pairs by fine-grained topics (Gao et al., 2025) like algebra, geometry, and so on. Within each subcategory, ranking data are generated by forming pairwise combinations of questions. Specifically, for any two questions $(q_i, q_j)$ within the same subcategory, we create the ordered pair $(e_i, e_j, r_{ij})$ and its symmetric counterpart $(e_j, e_i, 1 - r_{ij})$ to reduce positional bias, where $r_{ij}$ denotes to the ranking label: it equals to 1 if the question $i$'s difficulty $d_i$ exceeds the question $j$'s difficulty $d_j$, and 0 otherwise. To account for the influence of the magnitude gap between difficulty scores $d$, we assign a weight to each ranking pair. For $(e_j, e_j, r_{ij})$, the weight is $w_{ij} = \sqrt{|d_i - d_j|}$. We use these pairwise samples to train a multilayer perceptron (MLP) as the difficulty ranker. The input $\mathbf{x}$ of MLP is a fusion of absolute and relative features, which is concatenated as:

$$\mathbf{x}_{ij} = \big( e_i \,\|\, e_j \,\|\, e_i - e_j \,\|\, |e_i - e_j| \big)$$

Weighted binary cross-entropy loss is used as the optimization target to train an MLP with hyper-parameter $\phi$. The MLP ranker outputs the logit and then calculated $p(i, j)$, which measures the probability that question $i$ is more difficult than question $j$:

$$p(i, j) = \sigma(\mathrm{MLP}_\phi(\mathbf{x}_{ij}))$$

$$\mathcal{L}_{\mathrm{BCE}} = \frac{1}{|\mathcal{B}|} \sum_{(i,j) \in \mathcal{B}} w_{ij} \Big( - r_{ij} \log p(i, j) - (1 - r_{ij}) \log\big(1 - p(i, j)\big) \Big)$$

$p_{ij}$ will be used to form the difficulty reward for rewriting questions in reinforcement learning.

### 4.2 REINFORCEMENT LEARNING FOR DIFFICULTY EVOLUTION

We use Qwen3-8B (Team, 2025) as the base model and train a math question rewriting model via Supervised Fine-Tuning (SFT) and Reinforcement Learning (RL). The SFT stage primarily serves as a cold start, which equips the base model with an initial ability to rewrite original math questions $q$ into more difficult ones $q'$ and standardizes the output format. We curate $2,000$ high-quality rewritten questions $\mathcal{D}$ by distilling and filtering GPT-5-mini [2] outputs, and used this corpus to perform full-parameter fine-tuning of Qwen3-8B:

$$\theta_{\mathrm{SFT}} \leftarrow \arg\min_\theta \mathbb{E}_{(q,q') \in \mathcal{D}} \big[ - \log(\mathcal{P}_\theta(q' \mid q)) \big]$$

$\theta$ denotes to the original parameter of the base model and $\theta_{SFT}$ denotes to the new parameter after full-parameter fine-tuning.

In the RL stage, we use the Group Sequence Policy Optimization (GSPO) algorithm (Zheng et al., 2025) for optimization. Starting from SFT model with parameter $\theta_{\mathrm{SFT}}$, for each $q$ we sample a group $\mathcal{G}(q) = \{q'_k\}_{k=1}^K \sim \mathcal{P}_\theta(\cdot|q)$ with group size $K$, compute rewards $R(q, q'_k)$ and group-normalized advantages, and update model parameter $\theta$ with GSPO. Our rewards consist of:

- **Difficulty Reward.** The difficulty reward measures whether the rewritten question is more difficult than the original. We obtain embedding vectors $e$ and $e'$ for the original question $q$ and its rewrite $q'$, respectively, and concatenate them as the input to the difficulty ranker. The ranker returns $p(q', q)$ and its reverse $p(q, q')$, which respectively describe the probabilities that $q'$ is harder than $q$ and vice versa. The difficulty reward is defined as:

$$R_{\mathrm{diff}} = p(q', q) - p(q, q')$$

---

[1] https://platform.openai.com/docs/models/text-embedding-3-large
[2] https://openai.com/

- **Correctness Reward.** Due to the base model's limited reasoning ability, the rewritten questions may contain errors. We use a teacher model GPT-5-mini to verify each rewrite and output a correctness reward $R_{\text{cor}} \in \{-1, 1\}$, with $-1$ for incorrect and $1$ for correct.
- **Keyword and Length Reward.** To enforce a parsable format and penalize overly verbose rewrites, we add auxiliary keyword and length rewards $R_{\text{key}}$ with small coefficients to avoid overshadowing difficulty and correctness rewards.

Given reward weights $\alpha$, $\beta$, and $\gamma$, the final reward $R$ is defined as their weighted sum. For each $q$ with samples $\mathcal{G}(q) = \{q_k'\}_{k=1}^K$, let $R_k \equiv R(q, q_k')$ and compute advantage $A_k$:

$$R = \alpha R_{\text{diff}} + \beta R_{\text{cor}} + \gamma R_{\text{key}}, \qquad A_k = \frac{R_k - \frac{1}{K}\sum_{k=1}^K R_k}{\text{std}(R_{1:K})},$$

where $\text{std}(R_{1:K})$ denotes the standard deviation over the group $\{R_i\}_{i=1}^K$. Let $\theta_{\text{old}}$ be the policy for sampling $\mathcal{G}(q)$, initialized as $\theta_{\text{SFT}}$ and updated online to $\theta$. The sequence-level importance ratio:

$$\rho_k(\theta) = \left(\frac{\mathcal{P}_\theta(q_k' \mid q)}{\mathcal{P}_{\theta_{\text{old}}}(q_k' \mid q)}\right)^{1/|q_k'|} = \exp\left(\frac{1}{|q_k'|}\sum_{t=1}^{|q_k'|} \log \frac{\pi_\theta(q_{k,t}' \mid q, q_{k,<t}')}{\pi_{\theta_{\text{old}}}(q_{k,t}' \mid q, q_{k,<t}')}\right)$$

Here, $t$ indexes positions in the generated response $q_k'$; $q_{k,t}'$ denotes the $t$-th response token and $q_{k,<t}'$ is its prefix. The normalization length $|q_k'|$ counts response tokens only, and accordingly $\log \mathcal{P}_\theta(q_k' \mid q)$ is computed as the sum of token-level log probabilities over those response positions. We maximize the clipped surrogate:

$$\mathcal{L}_{\text{GSPO}}(\theta) = \mathbb{E}_{q \in \mathcal{D}}\left[\frac{1}{K}\sum_{k=1}^K \min\left(\rho_k(\theta)\,A_k,\ \text{clip}(\rho_k(\theta),\, 1-\varepsilon,\, 1+\varepsilon)\,A_k\right)\right]$$

At iteration $i$, we set $\theta_{\text{old}} = \theta^{(i)}$; for each source question $q$, we sample a group $\mathcal{G}(q)$ and compute the component rewards $R_{\text{diff}}$, $R_{\text{cor}}$, $R_{\text{key}}$ and their mixture $R$, form the group-normalized advantages $A_k$, and then update the parameters by maximizing the GSPO surrogate:

$$\theta^{(i+1)} = \arg\max_\theta \mathcal{L}_{\text{GSPO}}(\theta)$$

Once the reward converges, we obtain our final difficulty-aware math rewriting model **RIDE-8B**.

Considering that existing mathematical datasets may not provide complete solutions, we only take the original question $q_{\text{original}}$ and its answer $a_{\text{original}}$ from benchmarks or training data as input. The corresponding prompt is shown in Appendix B.3. Instead of prescribing specific rules, the prompt provides high-level guidance for increasing difficulty, allowing the model to freely rewrite questions, thereby creating unpredictable paths of difficulty progression. Subsequently, we take the prompt context as input into RIDE-8B, which infers the rewritten question $q_{\text{rewrite}}$, solution $s_{\text{rewrite}}$, and answer $a_{\text{rewrite}}$:

$$(q_{\text{original}}, a_{\text{original}}) \xrightarrow{\text{RIDE-8B}} (q_{\text{rewrite}}, s_{\text{rewrite}}, a_{\text{rewrite}}).$$

Including the solution $s_{\text{rewrite}}$ as a part of rewriting output not only strengthens the consistency between the rewritten question and its answer but also serves as a chain-of-thought signal, which can facilitate further training of models to improve their mathematical reasoning ability.

## 5 EXPERIMENT

### 5.1 EXPERIMENT SETUP

We use our rewriting model **RIDE-8B** to perform multi-round difficulty evolution on the competition-level mathematical-reasoning benchmarks AMC-23 and AIME-24, which contain 40 and 30 questions, respectively. Using GPT-5, we filter out questions that are unsolvable or have incorrect answers to construct the final perturbed benchmarks **RIDE-AMC** and **RIDE-AIME**, which are further manually checked to ensure answer accuracy, with annotators holding at least a bachelor's

degree. We evaluate the pass@1 performance of cutting-edge open-source and proprietary models in the zero-shot setting. In addition, we extract data from the DeepMath dataset for rewriting, and after filtering with DeepSeek-V3.1 (DeepSeek-AI, 2024), we obtain 10K high-quality mathematical reasoning samples, referred to as **RIDE-DeepMath**, which serve as training data to improve the models' ability for mathematical reasoning.

## 5.2 Main Results

We evaluate 23 LLMs comprising 8 proprietary models (including GPT-5 and Grok-4) and 15 open-source models (including DeepSeek-V3.1 and Kimi-K2-Instruct (Team et al., 2025)), spanning parameter counts from 0.6B to 1TB and covering multiple model series. We report pass@1 performance on the original AMC-23 and AIME-24 benchmarks, and then apply the same prompts to our perturbed benchmarks, RIDE-AMC and RIDE-AIME, again recording pass@1 performance. To quantify robustness, we compute the performance drop rate (PDR) (Li et al., 2024):

$$\text{PDR} = 1 - \frac{\frac{1}{|\mathcal{D}_a|} \sum_{(x,y) \in \mathcal{D}_a} \mathbb{I}[\text{LM}(x) = y]}{\frac{1}{|\mathcal{D}|} \sum_{(x,y) \in \mathcal{D}} \mathbb{I}[\text{LM}(x) = y]},$$

where $x$ and $y$ denote the question and the corresponding answer, respectively, $\mathcal{D}$ is the original benchmark (AMC-23 or AIME-24), $\mathcal{D}_a$ is the corresponding perturbed benchmark (RIDE-AMC or RIDE-AIME), $\text{LM}(\cdot)$ is the model's prediction, and $\mathbb{I}[\cdot]$ is the indicator function. Lower PDR indicates greater robustness. Results are presented in Table 1.

| Model | Params | AIME-24 | RIDE-AIME | PDR (↓%) | AMC-23 | RIDE-AMC | PDR (↓%) |
|---|---|---|---|---|---|---|---|
| *Proprietary Models* | | | | | | | |
| Claude-4-Sonnet | - | 40.00 | 36.67 | 8.32 | 85.00 | 82.50 | 2.94 |
| Claude-4.1-Opus | - | 53.33 | 36.67 | 31.24 | 87.50 | 82.50 | 5.71 |
| Gemini-2.0-Flash | - | 30.00 | 30.00 | 0.00 | 85.00 | 80.00 | 5.88 |
| Gemini-2.5-Pro | - | 90.00 | 90.00 | 0.00 | 100.00 | 100.00 | 0.00 |
| GPT-3.5-turbo | - | 10.00 | 3.33 | 66.70 | 55.00 | 50.00 | 9.09 |
| GPT-o3-mini | - | 76.67 | 76.67 | 0.00 | 100.00 | 97.50 | 2.50 |
| GPT-5 | - | 93.33 | 86.67 | 7.14 | 100.00 | 100.00 | 0.00 |
| Grok-4 | - | 100.00 | 96.67 | 3.33 | 100.00 | 95.00 | 5.00 |
| *Open-source Models* | | | | | | | |
| DeepSeek-V3.1 | 671B | 60.00 | 56.67 | 5.55 | 87.50 | 85.00 | 2.86 |
| DeepSeek-V3.1-Thinking | 671B | 93.33 | 83.33 | 10.71 | 100.00 | 100.00 | 0.00 |
| GLM-4.5 | 355B | 93.33 | 80.00 | 14.28 | 100.00 | 92.50 | 7.50 |
| Kimi-K2-Instruct | 1TB | 73.33 | 56.67 | 22.72 | 90.00 | 85.00 | 5.56 |
| Llama2-70B | 70B | 26.67 | 3.33 | 87.51 | 40.00 | 30.00 | 25.00 |
| Llama3.1-70B | 70B | 20.00 | 6.67 | 66.65 | 47.50 | 30.00 | 36.84 |
| Llama3.1-405B | 405B | 23.33 | 6.67 | 71.41 | 47.50 | 35.00 | 26.32 |
| Llama4-Scout | 109B | 30.00 | 3.33 | 88.90 | 57.50 | 45.00 | 21.74 |
| Mistral-large | 123B | 30.00 | 20.00 | 33.33 | 70.00 | 60.00 | 14.29 |
| Qwen2.5-7B | 7B | 10.00 | 10.00 | 0.00 | 50.00 | 30.00 | 40.00 |
| Qwen2.5-Math-7B | 7B | 16.67 | 3.33 | 80.02 | 60.00 | 32.50 | 45.83 |
| Qwen2.5-Math-72B | 72B | 26.67 | 10.00 | 62.50 | 70.00 | 60.00 | 14.29 |
| Qwen2.5-72B | 72B | 16.67 | 10.00 | 40.01 | 70.00 | 52.50 | 25.00 |
| Qwen3-0.6B | 0.6B | 10.00 | 3.33 | 66.70 | 37.50 | 20.00 | 46.67 |
| Qwen3-30B-A3B | 30B | 83.33 | 80.00 | 4.00 | 80.00 | 72.50 | 9.38 |

Table 1: Comparison of the model's pass@1 performance before and after benchmark perturbation. After benchmark perturbation, most models show a significant performance drop. The top-3 models with the largest average performance decline are highlighted in blue.

For AMC-23 and RIDE-AMC we use Qwen3's non-thinking mode, whereas for AIME-24 and RIDE-AIME we use Qwen3's thinking mode. Results show that our difficulty evolution perturbation method leads to performance drops in the vast majority of both proprietary and open-source models, even state-of-the-art models including GPT-5 and DeepSeek-V3.1-Thinking exhibit declines, which shows the validation of our method. On the more challenging benchmark AIME-24, the average PDR is higher than on AMC-23. Across all models, the mean PDR reached 21.73%. From the model perspective, proprietary models generally have lower PDRs, indicating better robustness. Open-source models showed substantial variation, and several LLaMA and Qwen series models suffered large declines, with the highest PDR reaching 88.90%.

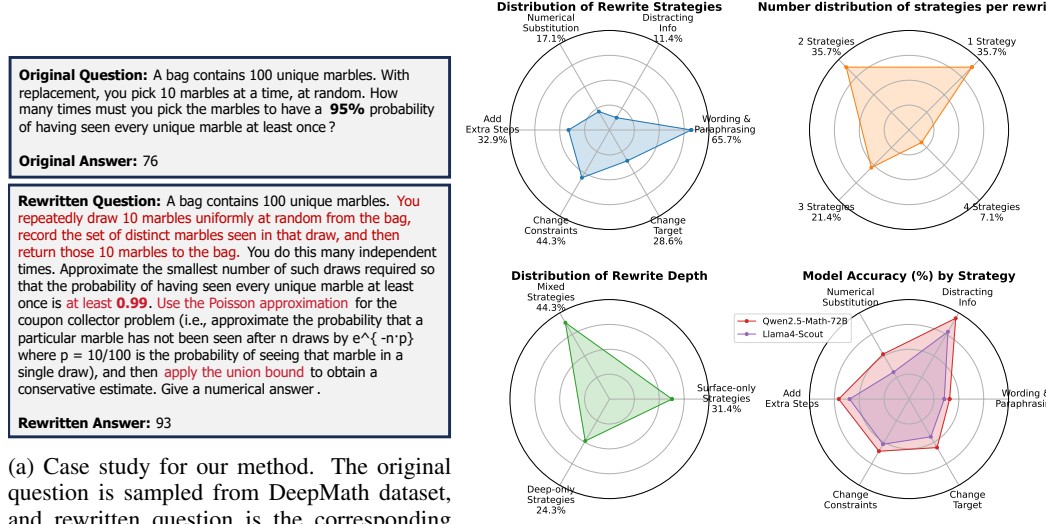

**Original Question:** A bag contains 100 unique marbles. With replacement, you pick 10 marbles at a time, at random. How many times must you pick the marbles to have a **95%** probability of having seen every unique marble at least once ?

**Original Answer:** 76

**Rewritten Question:** A bag contains 100 unique marbles. You repeatedly draw 10 marbles uniformly at random from the bag, record the set of distinct marbles seen in that draw, and then return those 10 marbles to the bag. You do this many independent times. Approximate the smallest number of such draws required so that the probability of having seen every unique marble at least once is at least **0.99**. Use the Poisson approximation for the coupon collector problem (i.e., approximate the probability that a particular marble has not been seen after n draws by $e^{-n \cdot p}$ where $p = 10/100$ is the probability of seeing that marble in a single draw), and then apply the union bound to obtain a conservative estimate. Give a numerical answer .

**Rewritten Answer:** 93

(a) Case study for our method. The original question is sampled from DeepMath dataset, and rewritten question is the corresponding rewritten version in RIDE-DeepMath.

(b) Rewrite strategy analysis.

Figure 3: Case study and analysis. (a) The original and rewritten questions from DeepMath and RIDE-DeepMath. (b) The analysis of rewrite strategies.

## 5.3 CASE STUDY AND REWRITE STRATEGY ANALYSIS.

Figure 3a illustrates the rewritten question. The original question is a classic coupon collector question with replacement. The rewritten question (1) clarifies the "pick 10 marbles at a time with replacement" procedure without changing the core idea; (2) raises the target success probability from 95% to 99%, increasing the required trials; and (3) requires using the Poisson approximation and the union bound, increasing restrictions and shifting the solution from exact combinatorial counting to multi-step reasoning. To systematically analyze these rewrites, we distill six core rewrite strategies from the questions in the RIDE-AIME and RIDE-AMC datasets: (1) Wording and Paraphrasing Modification, (2) Distracting Info, (3) Numerical Substitution, (4) Add Extra Steps, (5) Change Constraints, and (6) Change Target. (Detailed descriptions are provided in the Appendix B.4). Our analysis (Figure 3b) reveals that these strategies are frequently combined. Over 60% of the rewritten questions are composed of two or more strategies. To further investigate the impact on reasoning depth, we categorize strategies 1-3 as Surface Strategies, which have a minor impact on reasoning depth, and strategies 4-6 as Deep Strategies, which significantly affect the required reasoning. Statistics show that rewrites using a mix of both Surface and Deep strategies are the most common (44.3%). Finally, we evaluate the accuracy of Qwen2.5-Math-72B and Llama4-Scout on questions corresponding to each strategy, showing that performance varies across different strategies. In summary, our method can generate combinations of one or more rewrite strategies without explicitly defining specific rules. These combinations effectively provide a difficulty perturbation for the original question.

## 5.4 ANALYSIS

In this subsection, we will conduct more experiments to further explore and analyze the effectiveness and impact of our method.

**Win Rate Comparison of RIDE and Other methods.** To evaluate rewrite quality, we design prompts assessing completeness, clarity, conceptual consistency, structural perturbations, and avoidance of shortcuts (Appendix B.3). GPT-5 acts as an evaluator to compare the outputs from different methods. Each pair is judged twice with swapped order to mitigate position bias. We compute two win rate metrics: (1) Average win rate—the proportion of times a method is selected across both evaluations; (2) Consistent win rate—the proportion of questions where a method is selected in both evaluations. Ties are allowed in the evaluator's output. We run three rounds and report average results. We compare our method with the following method: (1) **Rule-based method.** Following GSM-Plus (Li et al., 2024), we use Qwen3-8B to generate perturbed AMC-23 and AIME-24 data

| Compared Method | Dataset | Average | | Consistent | |
|---|---|---|---|---|---|
| | | Baseline | RIDE | Baseline | RIDE |
| Rule-based | AMC-23 | 40.83 | **58.75** | 28.33 | **48.33** |
| | AIME-24 | 36.67 | **63.33** | 28.89 | **55.56** |
| Contrastive Prompting | AMC-23 | 49.17 | **50.83** | 40.00 | **40.83** |
| | AIME-24 | 32.78 | **67.22** | 26.67 | **61.11** |
| SFT-only | AMC-23 | 38.75 | **60.00** | 33.33 | **53.33** |
| | AIME-24 | 36.67 | **62.22** | 26.67 | **52.22** |
| w/o $R_{\text{diff}}$ | AMC-23 | 43.75 | **54.17** | 35.83 | **47.50** |
| | AIME-24 | 27.78 | **72.22** | 21.11 | **65.56** |

Table 2: Win rate comparison of RIDE and other methods. Baseline refers to the compared method.

with the same prompting strategy as rule-based baseline. (2) **Contrastive prompting.** Following the contrastive prompting method (Yao, 2025), we instruct Qwen3-8B to generate an easier and a more difficult rewrite at the same time and choose the difficult one as the final rewrite question. (3) **SFT-only model and RL model without difficulty reward.** We generate rewritten questions using the SFT-only model (without RL) and an RL model trained without the difficulty reward $R_{\text{diff}}$ as part of our ablation study. As shown in Table 2, our method significantly outperforms the rule-based and contrastive prompting on both metrics, demonstrating that RIDE rewrites achieve higher quality. Ablation study also shows the effect of our RL stage and difficulty reward.

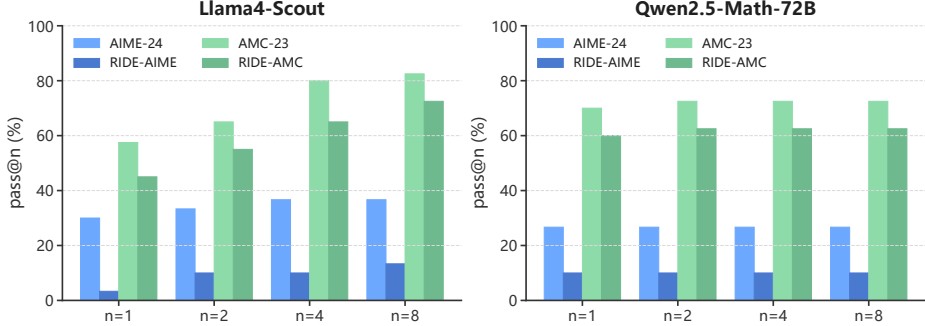

Figure 4: Comparison of pass@n performance before and after benchmark perturbation.

**Pass@n Performance.** To verify the stability of our method against model performance perturbations, we evaluate Qwen2.5-Math-72B and Llama-4-Scout on AIME-24, AMC-23, and their perturbed RIDE-AIME and RIDE-AMC using pass@n (n = 1, 2, 4, 8). Results (Figure 4) show that accuracy improves as n increases, but performance on perturbed benchmarks remains notably lower, confirming the stability of our method in testing robustness. Moreover, Qwen2.5-Math-72B exhibits greater stability, with a relatively small gap between its pass@1 and pass@8 scores.

**Data Augmentation For Training Data.** Our method also serves as an effective data augmentation strategy for mathematical reasoning. Using RIDE-8B, we applied difficulty-evolution rewriting to the DeepMath dataset and filtered incorrect samples with DeepSeek-V3.1, yielding **RIDE-DeepMath** with over 10K examples. Without SFT, we trained Qwen3-0.6B, Qwen3-1.7B and Gemma3-4B-It via reinforcement learning with GSPO, producing **RIDE-Qwen-0.6B-Thinking**, **RIDE-Qwen-1.7B** and **RIDE-Gemma-4B**. Results (Table 3) show that models trained with reinforcement learning on our data achieve improvements over their base models. In particular, RIDE-Qwen-0.6B-Thinking and RIDE-Gemma3-4B have surpassed GPT-4o and Qwen2.5-Math-72B, and are close to Qwen3-4B in the non-thinking mode.

**Effect of IRT models.** We compare different IRT models and data augmentation methods for question difficulty estimation. The 2-PL model adds a discrimination parameter to the 1-PL, while the 3-PL further introduces a guessing parameter. We randomly mask 20% of the response matrix for augmentation and model fitting, and evaluated performance using AUC-ROC. As shown in Table 4,

| Model | Params | AIME-25 |
|---|---|---|
| GPT-4o | - | 13.33 |
| Qwen3-0.6B | 0.6B | 3.33 |
| Qwen3-0.6B-Thinking | 0.6B | 10.00 |
| RIDE-Qwen-0.6B-Thinking | 0.6B | **20.00** |
| Qwen3-1.7B | 1.7B | 6.67 |
| RIDE-Qwen-1.7B | 1.7B | **13.33** |
| Qwen3-4B | 4B | 23.33 |
| Gemma3-4B-It | 4B | 13.33 |
| RIDE-Gemma3-4B | 4B | **20.00** |
| Qwen2.5-Math-7B | 7B | 6.67 |
| Qwen2.5-Math-72B | 72B | 13.33 |
| Qwen3-30B-A3B | 30B | 23.33 |

Table 3: Pass@1 performance comparison on AIME-25 benchmark. Models trained with reinforcement learning on our augmented rewritten data achieve higher performance than the corresponding base models.

| Model | Method | Data Augmentation | | AUC-ROC |
|---|---|---|---|---|
| | | VAE | Sampling | |
| 1-PL | VI | - | - | 89.81 |
| 1-PL | VI | ✓ | - | 91.01 |
| 1-PL | VI | - | ✓ | 91.06 |
| 1-PL | VI | ✓ | ✓ | **91.29** |
| 1-PL | MCMC | - | - | 89.81 |
| 2-PL | VI | - | - | 85.18 |
| 2-PL | VI | ✓ | ✓ | 86.56 |
| 3-PL | VI | - | - | 90.76 |
| 3-PL | VI | ✓ | ✓ | 91.01 |

Table 4: Comparison of different IRT models and methods. VI denotes Variational Inference and MCMC denotes Markov Chain Monte Carlo. We select the method combination with the highest AUC-ROC as IRT parameter estimation approach.

augmentation generally improves prediction accuracy, confirming its effectiveness. Given the high cost of MCMC, we ultimately adopt the 1-PL model with both augmentation methods and VI for parameter estimation, achieving a balance between accuracy and efficiency. Detailed results and the effect of the difficulty ranker are provided in Appendix C.2.

**Effect Analysis.** We additionally train RIDE-4B based on Qwen3-4B and keep SFT checkpoints (RIDE-4B-SFT, RIDE-8B-SFT) to disentangle SFT from RL effects. As shown in Table 5, we compute the Difficulty score using the predicted probability from our difficulty ranker, which indicates the likelihood of the rewritten question being more difficult than the original. We compare Qwen3-8B, the SFT-only RIDE-8B-SFT, and our RIDE-8B. The experiments demonstrate that the RL-tuned RIDE-8B achieves the highest average Difficulty score, thereby validating the effectiveness of our RL

| Model | Difficulty | Correctness |
|---|---|---|
| Qwen3-8B | 74.61 | 40.00 |
| RIDE-8B-SFT | 78.78 | 40.00 |
| RIDE-8B | **80.18** | **42.50** |
| RIDE-4B | - | **45.00** |
| RIDE-4B-DeepSeek | - | 37.50 |
| RIDE-4B w/o $R_{cor}$ | - | 32.50 |

Table 5: Effect analysis of the RL stage and correctness reward.

strategy. Additionally, we conducted ablation studies on the RIDE-4B model. We introduced two variations: (1) replacing the verifier model for the correctness reward $R_{cor}$ with DeepSeek-V3.2 (denoted as RIDE-4B-DeepSeek), and (2) removing $R_{cor}$ entirely. We employed GPT-5 to evaluate the correctness of the rewritten questions generated by these models. The results indicate that the standard RIDE-4B, which utilizes GPT-5-mini as the verifier, achieves the highest correctness. The RIDE-4B-DeepSeek model follows, as DeepSeek-V3.2 tends to output more false negatives (see Appendix C.3), whereas removing the correctness reward leads to a significant decline in correctness. This confirms the validity and necessity of our correctness reward.

**Statistics of Different Rewriting Models.** We train RIDE-1.7B/4B based on Qwen3-1.7B/4B and retain the SFT checkpoints, conducting statistical analyses across generation time, token count, correctness, and semantic similarity. The results demonstrate that both the SFT and RL stages yield effective improvements. Detailed experimental results and analyses are provided in Appendix C.4.

# 6 CONCLUSION

In this paper, we propose a question rewriting framework based on difficulty evolution. The method estimates question difficulty using IRT, then trains a difficulty ranker to provide a difficulty reward signal, guiding reinforcement learning for the rewriting model. Experiments show that our method causes significant performance drops in 26 LLMs, validating its effectiveness and robustness in adversarial perturbation. Our method outperforms traditional rule-based perturbations and can be applied for training data augmentation. Further analyses confirm its effectiveness, offering a new perspective on robust evaluation from the view of "question difficulty".

## ETHICS STATEMENT

All data used in this work are from publicly available open source datasets under their respective licenses. For IRT modeling, we rely exclusively on responses generated by LLMs, and no real student data or personally identifiable information is used. Our method is designed only to rewrite mathematical questions and does not attempt to infer or profile individuals.

## REPRODUCIBILITY STATEMENT

All our source code will be submitted in the Supplementary Materials, along with an explanation to facilitate reviewers in reproducing our work. The hyperparameter setup is introduced in Appendix B.2. We will also open source our mathematical question rewriting models RIDE-4B and RIDE-8B, our perturbation benchmarks RIDE-AMC and RIDE-AIME, and our augmented training data RIDE-DeepMath.

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

# A  THE USE OF LLMS

In the experimental phase, LLMs are mainly used in the following aspects:

- **As base models**: For example, our question rewriting model RIDE-8B is training upon Qwen3-8B.

- **As data filter**: We employ GPT-5 and DeepSeek-V3.1 to filter out questions that are unsolvable or have incorrect answers.

- **As evaluators**: During the reinforcement learning stage, we use GPT-5-mini to provide correctness rewards. Besides, GPT-5 serves as a teacher model to judge the win rate between two rewritten questions.

- **As evaluated models**: We evaluate the feasibility of our proposed method for difficulty-evolution robust perturbations on 26 LLMs.

In the non-experimental phase, LLMs are applied, including:

- **Code assistance**: Using GPT-5 for environment setup, code debugging, and partial script development. The LLMs-related code has been rigorously reviewed by humans.

- **File processing**: Using GPT-5 to handle files in formats such as JSON, and summarize some results in training logs.

- **Paper writing support**: Leveraging GPT-5 for grammar checking and polishing to improve accuracy and quality of academic writing.

# B  EXPERIMENTAL SETUP

## B.1  IMPLEMENTATION DETAILS

We use the API interface provided by OpenAI to call models and obtain the response matrix of LLMs (for the GLM series, we use the ZHIPU interface). The size of the response matrix is 35×2000. We perform IRT parameter estimation through *py-irt* (Lalor & Rodriguez, 2023). The statistics of each student model's performance is shown in Figure 5, along with the accuracy and estimated ability $\theta$ are shown in Table 6. The results indicate that within the same model series (e.g., Qwen2.5 or Qwen3), model parameter size is positively correlated with capability. However, this relationship is not numerically linear. Additionally, frontier reasoning models generally exhibit higher capability scores.

In the SFT stage, we use the *LLaMA-Factory* (Zheng et al., 2024) framework for training, with a dataset of $2,000$ rewritten mathematical questions distilled from GPT-5-mini. In the reinforcement learning stage, we use the *TRL* (von Werra et al., 2020) framework for training, with $3,000$ questions extracted from the DeepMath dataset (He et al., 2025). In our experiments, model performance improved after benchmark perturbations in a small number of cases (Table 7), which is more common with Qwen3 in thinking mode, which further confirms its strong robustness and strong reasoning ability.

## B.2  HYPERPARAMETER SETUP

The hyperparameters used in our work are summarized in Table 8. For the IRT parameter estimation and the training of the difficulty ranker, we perform a greedy search to select the optimal group of hyperparameters based on evaluation metrics. For the SFT and reinforcement learning stages, the hyperparameters are chosen by balancing model performance with the available computational resources.

## B.3  PROMPT STRATEGY

For constructing the response matrix and evaluating model performance on both the original benchmark and the perturbed benchmark, we use the same zero-shot prompt to instruct models to answer

| Model | Parameter | Pass@1 (%) | $\theta$ |
|---|---|---|---|
| Claude-3.5-haku | - | 46.75 | -1.132 |
| Claude-3.7-sonnet | - | 76.85 | 0.710 |
| Deepseek-R1-Distill-Llama-8B | 8B | 74.65 | 0.552 |
| Deepseek-R1-Distill-Qwen-1.5B | 1.5B | 71.65 | 0.324 |
| Deepseek-R1-distill-Qwen-14B | 14B | 88.75 | 1.822 |
| Deepseek-v3-0328 | 671B | 85.85 | 1.534 |
| Gemini-2.0-flash | - | 82.65 | 1.203 |
| Gemini-2.5-flash | - | 93.15 | 2.406 |
| Gemma3-12B-it | 12B | 64.80 | -0.115 |
| Gemma3-4B-it | 4B | 49.80 | -0.970 |
| GLM-4-9B-chat | 9B | 30.50 | -2.038 |
| GLM-4-plus | - | 60.25 | -0.347 |
| GLM-Z1-airx | - | 93.10 | 2.417 |
| GLM-Z1-flash | - | 91.80 | 2.222 |
| GPT-3.5-turbo | - | 29.85 | -2.079 |
| GPT-4o | - | 64.15 | -0.135 |
| Grok-2 | - | 62.35 | -0.213 |
| Grok-3 | - | 81.25 | 1.064 |
| Kimi-K2-Instruct | 1TB | 93.05 | 2.462 |
| Llama-3.1-70B | 70B | 53.10 | -0.770 |
| Llama-3.1-8B | 8B | 28.20 | -2.168 |
| Llama-4-scout | 109B | 70.90 | 0.284 |
| Mistral-7B-Instruct-V0.2 | 7B | 13.20 | -3.427 |
| Mistral-Small-3.1-24B-Instruct | 24B | 50.05 | -0.947 |
| Qwen-math-72B | 72B | 75.05 | 0.555 |
| Qwen-math-7B | 7B | 63.60 | -0.143 |
| Qwen2.5-0.5B | 0.5B | 19.25 | -2.799 |
| Qwen2.5-14B | 14B | 64.90 | -0.106 |
| Qwen2.5-72B | 72B | 71.90 | 0.332 |
| Qwen2.5-7B | 7B | 56.20 | -0.541 |
| Qwen3-0.6B | 0.6B | 26.95 | -2.263 |
| Qwen3-14B | 14B | 77.05 | 0.735 |
| Qwen3-32B | 32B | 79.00 | 0.883 |
| Qwen3-235B-A22B | 235B | 83.95 | 1.262 |
| QwQ-32B | 32B | 95.00 | 2.813 |

Table 6: Statistics of each student model's performance and estimated ability.

| Model | Params | AIME-24 | RIDE-AIME | PDR ($\downarrow$%) | AMC-23 | RIDE-AMC | PDR ($\downarrow$%) |
|---|---|---|---|---|---|---|---|
| | | *Proprietary Models* | | | | | |
| Grok-3 | - | 60.00 | 63.33 | ↑5.55 | 92.50 | 87.50 | 5.41 |
| | | *Open-source Models* | | | | | |
| Qwen3-14B | 14B | 73.33 | 76.67 | ↑4.55 | 80.00 | 70.00 | 12.50 |
| Qwen3-235B-A22B | 235B | 90.00 | 93.33 | ↑3.70 | 97.50 | 95.00 | 2.56 |

Table 7: Cases of performance improvement on perturbed benchmarks.

mathematical questions (Figure 6), requiring only that answers follow a specific format for verification. In practice, we find that even without explicitly telling the model to "think step by step", all evaluated models still produce chain-of-thought outputs.

Figure 7 shows the prompt we use to guide the model in rewriting mathematical questions during the SFT and reinforcement learning stages. We do not impose a single, detailed rule. Instead, we briefly outline several ways to increase difficulty and allow the model to improvise, making the trajectory of difficulty evolution unpredictable.

Figure 8 presents the prompt we use to instruct GPT-5 to choose the one that better meets the requirements from two rewritten questions, enabling us to compute the win rate of our method against the rule-based approach. We specify in detail that the rewritten question must address covering question completeness, clarity, conceptual consistency, structural perturbations, and the avoidance of reasoning shortcuts.

| Hyperparameter | Description | Value |
|---|---|---|
| *IRT & Difficulty Ranker* | | |
| $M$ | The number of student models | 35 |
| $N$ | The number of mathematical questions | 2,000 |
| $S$ | The number of virtual students by VAE | 200 |
| $T$ | The number of virtual students by Sampling | 200 |
| $\beta$-VAE | Weight of KLD loss for VAE | 0.5 |
| Latent dimension | Dimension of latent space | 32 |
| Priors | Prior distribution of IRT parameters | vague |
| Random Seed | Random seed for partitioning missing values in the response matrix | 42 |
| Batch size | Number of samples per batch | 64 |
| Embedding dimension | Dimension of input embeddings | 3,072 |
| Hidden dimension | Hidden layer size of MLP | (512,256) |
| Dropout rate | Dropout probability for hidden layers | 0.3 |
| Activation | Non-linear activation function | ReLU |
| Epochs (CV) | Training epochs for cross-validation | 8 (5-CV) |
| *Supervised Fine-Tuning* | | |
| Device | Device used for SFT | 1 * NVIDIA A800 80GB PCIe |
| Finetuning type | Finetuning strategy | full |
| Template | Prompt/format template of LLaMA-Factory | qwen3 |
| Cutoff length | Max sequence length (tokens) | 8192 |
| Per-device train batch size | Batch size per device | 2 |
| Gradient accumulation steps | Grad accumulation steps | 4 |
| Learning rate | Initial learning rate | 1e-5 |
| Epochs | Number of training epochs | 3.0 |
| Warmup ratio | Warmup ratio | 0.1 |
| Bf16 | bfloat16 training | true |
| Deepspeed (Rasley et al., 2020) | Deepspeed Stage | 3 |
| *Reinforcement Learning* | | |
| Device | Device used for SFT | 4 * NVIDIA A800 80GB PCIe |
| Finetuning type | Finetuning strategy | LoRA |
| Method | RL algorithm | GSPO |
| Per-device batch size | Prompts per device per step | 1 |
| Gradient accumulation steps | Steps to accumulate gradients | 4 |
| Group size $K$ | Generations per prompt | 4 |
| Max prompt length | Max tokens of prompt | 512 |
| Max new tokens | Max tokens to generate per sample | 4096 |
| Temperature | Sampling temperature | 0.7 |
| Top-p | Nucleus sampling p | 0.95 |
| Max steps | Maximum training steps | 500 |
| KL coef | KL penalty coefficient | 0.0 |
| LoRA rank | Rank | 64 |
| LoRA $\alpha$ | Scaling factor | 64 |
| LoRA dropout | Dropout probability | 0.05 |
| Deepspeed | Deepspeed Stage | 2 |
| $\alpha$ | Weight of difficulty reward | 0.5 |
| $\beta$ | Weight of correctness reward | 0.5 |
| $\gamma$ | Weight of keyword and length reward | 0.3 |

Table 8: Main hyperparameter setup.

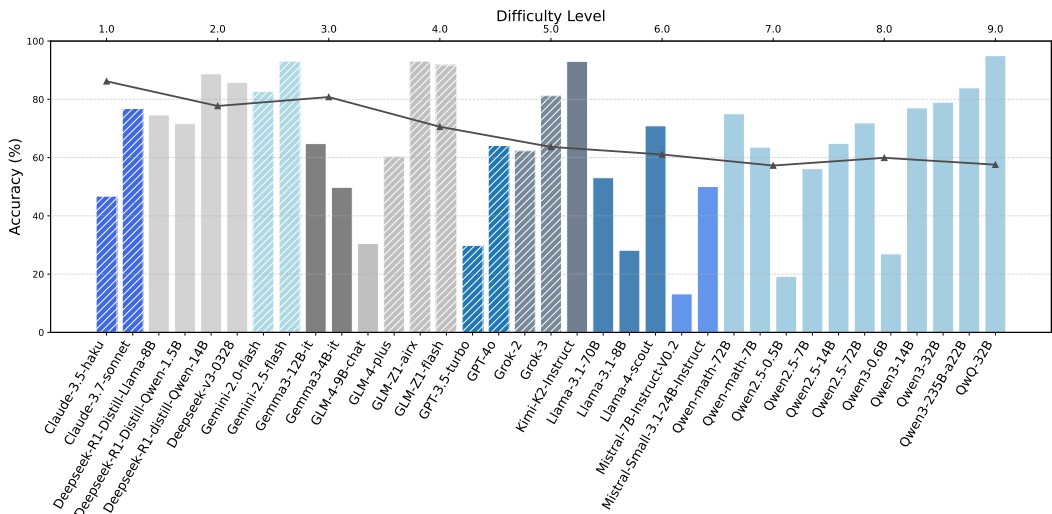

Figure 5: Statistics of each student model's performance. Difficulty Level is given by DeepMath Dataset.

---

**Instructions:** You are a mathematical reasoning assistant. Now solve the following question. The final answer should be marked with \\boxed{}.
{QUESTION}

---

Figure 6: Prompt for mathematical question answering.

## B.4 REWRITE STRATEGY

We distill six core rewrite strategies from the rewritten questions: (1) Wording and Paraphrasing Modification, (2) Distracting Info, (3) Numerical Substitution, (4) Add Extra Steps, (5) Change Constraints, and (6) Change Target. The detailed descriptions are shown in Table 9. To determine the strategies involved in each question, we employed a model-based annotation process. We fed each question to both GPT-5 and DeepSeek-V3.2 three times each, for a total of six annotations. A specific strategy was only recorded as being present for a given question if it was selected by a majority (i.e., at least three out of six) of the model runs.

| Rewrite Strategy | Definition |
|---|---|
| Wording & Paraphrasing | Rephrases the question using different vocabulary, sentence structures, or symbols without altering the mathematical conditions or numerical values. |
| Distracting Info | Adds extra information, definitions, or numerical values to the question description that are irrelevant to the solution. |
| Numerical Substitution | Replaces one or more key numerical values in the question statement with new values. |
| Add Extra Steps | Introduces new events, steps, or physical quantities into the original question scenario, which must be incorporated into the calculation. |
| Change Constraints | Adds new, stricter limitations or assumptions to the original question, narrowing the sample space, or alters the core rules to change the solution's strategy space or domain. |
| Change Target | The question's context and variable-solving process may remain the same, but the final algebraic expression to be calculated is replaced with a more complex target or is abstracted into parameters to explore a more general solution. |

Table 9: Definitions of the Rewrite Strategies.

**Instructions:** Please rewrite the following math problem into a significantly more difficult version, such that the likelihood of a model answering it correctly decreases.
When rewriting:
1. Question difficulty can be increased by adding more calculations, implicit conditions, higher-level math concepts, or distracting/unnecessary information.
2. You must not increase difficulty by adding multiple sub-questions or turning the problem into a proof question. The rewritten problem must remain a single well-defined question.
3. Solution steps must be complete and logically sound, but kept concise and relatively short (avoid excessively long derivations).
4. Answer must be unique and consistent with the solution steps.
5. Output must strictly follow this format:
<question>Rewritten question text</question>
<solution>Rewritten solution steps</solution>
<answer>Rewritten final answer</answer>
6. Do not omit any part, DO NOT output any other content except rewritten question, solution and answer.

Original Question
{QUESTION}
Original Answer
{ANSWER}

Figure 7: Prompt for rewriting mathematical questions.

**Instructions:** You are a rigorous math-problem rewrite judge. Your job is to compare two rewrites (A: model-generated, B: rule-generated) of the same original math problem and decide which rewrite is better for robustness testing. Do not solve any problem. Think silently and return the final preference.
# Goal
Select the rewrite that best preserves the core skill being tested while introducing meaningful structural perturbations (for robustness), with a problem that is clear, consistent, and fully solvable (preferably single-solution), and with difficulty ≥ the original (no shortcut reasoning).
# Required Criteria (all must be considered)
1) Completeness & Solvability: The rewritten problem must be fully specified, mathematically consistent, and solvable, preferably with a single correct solution.
2) Core-Concept Consistency: The rewrite must keep the core concept/skill of the original unchanged; numbers/labels/secondary conditions may change as long as the core assessment target is equivalent.
3) Clarity & No Leakage: Language must be clear, unambiguous, error-free, with no leaked conclusions, intermediate results, or hints that reduce the intended reasoning burden.
4) Structural Perturbation: Pure surface-level paraphrase without structural disturbance (e.g., only a few synonyms) is not a good rewrite for robustness; prefer changes in structure/order/representation that do not alter the core skill or answer.
5) Difficulty Constraint: The new problem's difficulty must be ≥ the original; it must not introduce shortcuts or reduce the required reasoning depth.
# Forbidden (strongly penalize; may make a candidate lose)
- Changing the problem type, core concept/skill, or final answer relative to the original.
- Inconsistent or conflicting data/units/constraints, or introducing multi-solution/unsolvable setups unless the original had that intent.
- Any answer leakage or revealing key intermediate results.
- Only superficial paraphrasing with no structural perturbation.
- Introducing reasoning shortcuts that lower difficulty.
# Tie-breaking priorities
When deciding between A and B:
1) Prioritize Completeness & Solvability and Core-Concept Consistency above all.
2) Next, prefer the one with meaningful structural perturbation and no leakage.
3) Next, prefer difficulty ≥ original without shortcuts.
4) If both are comparable across all criteria, output a tie.
# Process (follow strictly)
- Do not solve the problem or output any computation, steps, or answers.
- Analyze A and B against the criteria and forbidden list.
- Decide the winner using the tie-breaking priorities.
# Inputs
- Original question:
{ORIGINAL_QUESTION}
- Rewrite A:
{REWRITE_A}
- Rewrite B:
{REWRITE_B}

Figure 8: Prompt for judging which rewritten question is better.

## C  Effect Analysis

### C.1  Orthogonality between question difficulty and robustness verification

The verification of model robustness and the evolution of question difficulty can coexist in our method. We define the robustness of LLMs in mathematical reasoning as follows: for a question $x$ and the model's response correctness $y \in \{0, 1\}$, given a semantic-preserving transformation family $\mathcal{T}$, if $f(x) = y$, then for the vast majority of $t \in \mathcal{T}$, $f(t(x)) = y$ holds. Figure 10 shows that the semantic similarity between our rewritten and original questions consistently exceeds 80%, confirming that performance degradation does not stem from semantic disparity. Building on this, we decouple question difficulty from robustness verification. As detailed in Section 5.3, we identify "surface-level modifications" that preserve reasoning depth. An example is shown in Figure 9. In this example, the evolution of the question is manifested in the formalized mathematical condi-

| Original Question: In a table tennis tournament every participant played every other participant exactly once. Although there were twice as many right-handed players as left-handed players, the number of games won by left-handed players was 40% more than the number of games won by right-handed players. (There were no ties and no ambidextrous players.) What is the total number of games played? 
 Original Answer: 36 | Rewritten Question: In a round-robin table tennis tournament, every participant plays every other participant exactly once (no ties, no ambidextrous players). Let R be the number of right-handed players and L be the number of left-handed players, with R = 2L. Over the whole tournament, the total number of games won by left-handed players is 40% more than the total number of games won by right-handed players. Find the total number of games played. 
 Rewritten Answer: 36 |
|---|---|

Figure 9: A case study of the rewritten questions with surface-only strategies.

tions and changes in phrasing, which do not impact the depth of reasoning required, making this a case for verifying model robustness. For example, Qwen2.5-72B answered the original question correctly but failed on the rewritten question. We collected similar surface-only rewrites and categorized them into three difficulty levels—easy, medium, and hard—based on the accuracy rates of 10 LLMs on the original questions. The questions rewritten using the surface-only strategy maintain a reasoning depth consistent with the original questions, leaving the surface form as the sole variable. If the model's performance degradation were solely dependent on difficulty, the magnitude of the performance drop on the rewritten questions should vary significantly across the three difficulty levels. Conversely, if question difficulty and robustness verification can be decoupled, the variations in performance decline across the three difficulty levels should be insignificant. We evaluated the performance degradation between the original and rewritten questions across different difficulty levels (Table 10). The results indicate that surface-level rewriting induces performance degradation

| Level | Original Accuracy | Rewritten Accuracy | $\Delta$ |
|---|---|---|---|
| Easy | 80.00 | 68.57 | 11.43 |
| Medium | 68.57 | 55.71 | 12.86 |
| Hard | 62.50 | 43.75 | 18.75 |

Table 10: Performance drop comparison across different difficulty levels.

even on easy questions, thereby highlighting the necessity of robustness verification. Moreover, the performance drop across difficulty levels is not significant, with a particularly negligible variation in the magnitude of change between easy and medium questions. We additionally recorded the counts of "Correct on Original, Incorrect on Rewritten" and "Incorrect on Original, Correct on Rewritten" for each paired sample in the surface-level rewriting dataset and calculated the p-value using McNemar's test. The resulting p-value of $2.3847 \times 10^{-11}$ is substantially lower than 0.05, revealing a significant asymmetry. This demonstrates that the performance decline originates from the model's non-robustness rather than changes in difficulty. In summary, through case study and statistical analysis, we decoupled question difficulty from robustness verification and found that the two are orthogonal, allowing them to coexist within our method.

### C.2  Effect of IRT models and difficulty ranker.

As described in Section 5.4, we conducted additional experiments to verify the effectiveness of the choice of the IRT model and data augmentation methods. The results are shown in Table 11. We

consider two priors provided by *py-irt*: a vague prior that exerts minimal influence so the likelihood largely drives estimation, and a hierarchical prior that treats parameters as draws from a shared group-level distribution with hyperpriors, enabling partial pooling to stabilize estimates and shrink extremes, especially with sparse data. For evaluation, in addition to AUC-ROC for discrimination, we also report the Brier Score, the mean squared difference between predicted probabilities and observed outcomes (lower is better), which captures overall probabilistic accuracy and calibration and thus complements AUC-ROC's ranking focus.

| Model | Method | Data Augmentation | | Prior | AUC-ROC | Brier Score↓ |
|-------|--------|------|----------|-------|---------|-------------|
| | | VAE | Sampling | | | |
| 1-PL | VI | - | - | vague | 89.81 | 11.97 |
| 1-PL | VI | - | - | hierarchical | 90.03 | 11.89 |
| 1-PL | VI | ✓ | - | vague | 91.01 | 11.54 |
| 1-PL | VI | ✓ | - | hierarchical | 90.95 | 11.61 |
| 1-PL | VI | - | ✓ | vague | 91.06 | 11.25 |
| 1-PL | VI | - | ✓ | hierarchical | 91.03 | 11.29 |
| 1-PL | VI | ✓ | ✓ | vague | **91.29** | **11.29** |
| 1-PL | VI | ✓ | ✓ | hierarchical | 91.27 | 11.32 |
| 1-PL | MCMC | - | - | vague | 89.81 | 11.95 |
| 1-PL | MCMC | - | - | hierarchical | 90.08 | 11.85 |
| 2-PL | VI | - | - | vague | 85.18 | 15.87 |
| 2-PL | VI | - | - | hierarchical | 90.41 | 11.72 |
| 2-PL | VI | ✓ | - | vague | 86.19 | 15.50 |
| 2-PL | VI | ✓ | - | hierarchical | 90.89 | 11.64 |
| 2-PL | VI | - | ✓ | vague | 86.24 | 15.73 |
| 2-PL | VI | - | ✓ | hierarchical | 89.62 | 12.53 |
| 2-PL | VI | ✓ | ✓ | vague | 86.56 | 15.52 |
| 2-PL | VI | ✓ | ✓ | hierarchical | 89.81 | 12.27 |
| 2-PL | MCMC | - | - | vague | 84.56 | 15.72 |
| 2-PL | MCMC | - | - | hierarchical | 88.06 | 13.34 |
| 3-PL | VI | - | - | vague | 89.52 | 12.21 |
| 3-PL | VI | - | - | hierarchical | 89.52 | 12.21 |
| 3-PL | VI | ✓ | - | vague | 90.76 | 11.73 |
| 3-PL | VI | ✓ | - | hierarchical | 90.76 | 11.73 |
| 3-PL | VI | - | ✓ | vague | 90.05 | 12.50 |
| 3-PL | VI | - | ✓ | hierarchical | 90.05 | 12.50 |
| 3-PL | VI | ✓ | ✓ | vague | 91.01 | 12.13 |
| 3-PL | VI | ✓ | ✓ | hierarchical | 91.01 | 12.13 |

Table 11: Comparison of IRT models, data augmentation and prior method.

Besides, we explore regression, classification, and ranking methods to fit IRT-estimated question difficulties from textual features. Regression task is evaluated by RMSE, consistently yields values above 1 across models and embeddings, indicating large bias. Classification treats questions with difficulty below 0 as "easy" and above 0 as "hard," but F1 scores remain below 60% due to class imbalance. Finally, we adopt a pairwise ranking approach with AUC-ROC as the metric, achieving around 72%, with little variation across embedding models.

## C.3 Effect of LLM verifier

We use the proprietary GPT-5 series models to provide correctness rewards and as filtering models. To verify that our method can also use low-cost, open-source models, we employ GPT-5-mini and Deepseek-V3.2 (non-thinking mode) as verifiers. We provide $2,000$ math questions (without answers), along with the solutions and answers generated by two test models (Qwen2.5-Math-7B and Gemma3-12B), as input to these verifiers to judge the correctness of the test models' responses. The results, shown in Table 13, indicate that the verification accuracy of the closed-source model GPT-5-mini reach over 85%. In contrast, the accuracy of DeepSeek-V3.2 was relatively lower, but still over 65%. However, both verifiers achieve a precision of around 90%. This indicates that both models can precisely assign low reward signals to incorrect solutions and are effective for filtering erroneous data. The relatively low recall, however, implies that the verifiers tend to be conservative,

| Method | Task | Embedding | Metrics |
|---|---|---|---|
| LR | Regression | Qwen3-8B | 1.0375 |
| LR | Regression | BGE-M3 (Chen et al., 2024) | 1.1380 |
| SVR | Regression | Qwen3-8B | 1.0108 |
| MLP | Regression | Qwen3-8B | 1.0931 |
| Longformer (Beltagy et al., 2020) | Regression | Qwen3-8B | 1.0746 |
| SVM | Classification | Qwen3-8B | 58.34 |
| MLP | Classification | Qwen3-8B | 53.60 |
| MLP | Ranking | Qwen3-8B | 71.95 |
| MLP | Ranking | text-embedding-3-large | 71.98 |

Table 12: Comparison of regression, classification and ranking tasks on different models.

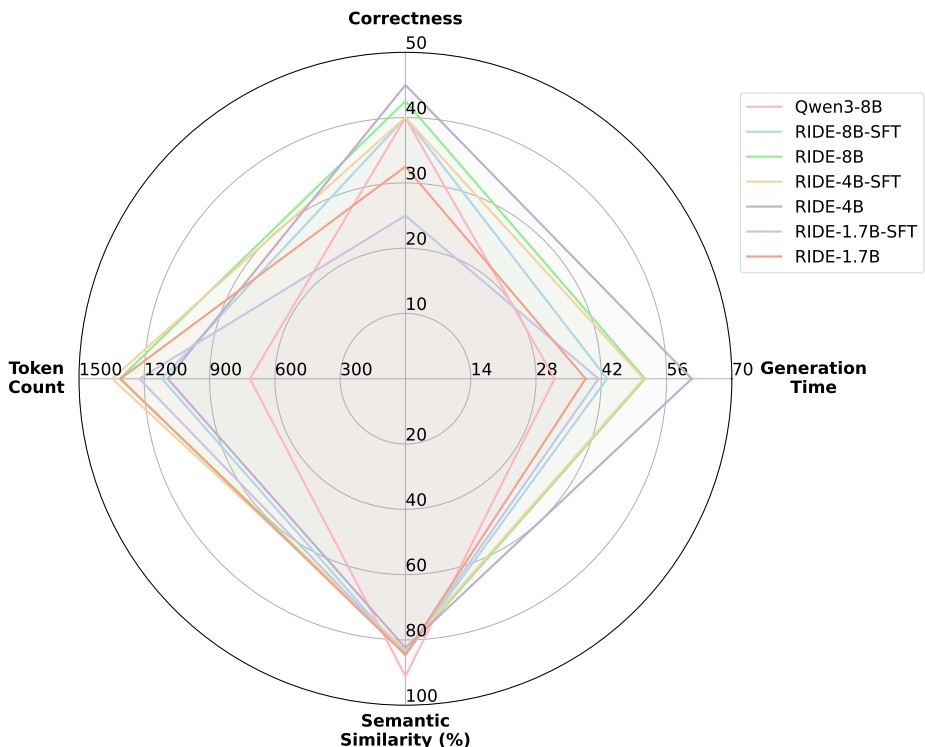

Figure 10: Statistics of average token count, average generation time, correctness and semantic similarity across different models.

flagging an otherwise correct solution and answer as incorrect. This may be because the model's solution contains minor errors even when the final answer is correct. This situation is acceptable, given that rule-based or model-based rewards in Reinforcement Learning with Verified Rewards (RLVR) also exhibit a high number of false negatives (Xu et al., 2025). While this situation affects training efficiency, it does not directly lead to training failure, as our verifier assigns low reward signals to incorrect rewrites. Therefore, our method can also use open-source models as verifiers and is not merely a distillation from proprietary models.

| Response Source | Verifier | Precision | Recall | $F_1$ | Accuracy |
|---|---|---|---|---|---|
| Qwen2.5-Math-7B | GPT-5-mini | 97.96 | 79.17 | 87.57 | 85.70 |
| | Deepseek-V3.2 | 89.48 | 62.19 | 73.38 | 71.30 |
| Gemma3-12B | GPT-5-mini | 98.38 | 79.71 | 88.06 | 86.00 |
| | Deepseek-V3.2 | 93.84 | 54.09 | 68.62 | 67.95 |

Table 13: Verification performance of GPT-5-mini and Deepseek-V3.2 verifiers on solutions generated by different response models.

### C.4 STATISTICS OF RIDE REWRITTEN MODEL.

Starting from Qwen3-1.7B and Qwen3-4B, we trained RIDE-1.7B and RIDE-4B, and preserved SFT checkpoints (RIDE-1.7B-SFT, RIDE-4B-SFT, RIDE-8B-SFT) to separate SFT from RL effects. The questions rewritten by each model are evaluated in terms of average generation time, average token count, correctness, and semantic similarity. Generation time and token count use the minimum of three runs, correctness is judged by GPT-5 and similarity is measured via text-embedding-3-large. The results are shown in Figure 10. Compared to Qwen3-8B, our SFT and reinforcement learning models produce longer outputs and slower inference, showing that rewrites expand questions, solutions, and answers. Semantic similarity drops slightly, indicating greater diversity while preserving affinity with the original questions. Reinforcement learning improves correctness over both the original and SFT-only models. RIDE-8B and RIDE-4B demonstrate much higher correctness than RIDE-1.7B, indicating that parameter size influences the correctness of generated questions. However, this difference is not significant between the 4B and 8B models, suggesting a limited effect of scale within this range.

## D    ITEM RESPONSE THEORY ANALYSIS

### D.1    HUMAN DIFFICULTY & LLM DIFFICULTY

In human cognition, the difficulty of a math question is often determined by its required reasoning depth. Consequently, researchers have established the AoPS (Art of Problem Solving) standard to measure the difficulty of math competition questions from a human perspective, and methods have emerged that use LLMs in conjunction with the AoPS standard to automatically assign difficulty ratings to math questions  (Gao et al., 2025). The DeepMath dataset, which we utilize in the IRT modeling phase, also applies this standard to provide difficulty levels for its questions. However, we discover a discrepancy between this human-centric standard and actual LLM performance. After grouping the responses of our 35 models on 2,000 questions by their given AoPS difficulty levels, we calculate the mean empirical accuracy for each level. The Spearman and Kendall correlation coefficients between the AoPS difficulty and the mean empirical accuracy are found to be **-0.5152** and **-0.5111**, respectively, indicating that while the expected negative correlation exists (i.e., higher human-perceived difficulty corresponds to lower model accuracy), the relationship is not sufficiently strong to consider them equivalent. Therefore, we conclude that a deviation exists between the human cognitive difficulty defined by the AoPS standard and the empirical error rate demonstrated by LLMs. Furthermore, treating lower empirical accuracy directly as higher "LLM difficulty" introduces a fundamental flaw: empirical accuracy assigns equal weight to the responses of all tested LLMs. We therefore introduce IRT to define difficulty, which allows us to derive a statistically robust difficulty value by incorporating the ability of all participating LLMs into the calculation.

Additionally, we compare the correlation of both the empirical error rate (ungrouped) and IRT difficulty against the human-perceived difficulty, again using Spearman and Kendall coefficients, as shown in Table 14. The results show that both empirical error rate and our IRT difficulty (fitted on 35 model responses) have a similar correlation with the given human difficulty. Our work is primarily focused on this "LLM difficulty" rather than human-perceived difficulty.

| vs Human-Rated Difficulty | Spearman | Kendall |
|---|---|---|
| Empirical Error Rate | 0.3362 | 0.2470 |
| IRT Difficulty | 0.3418 | 0.2468 |
| IRT Difficulty w/o augmentation | 0.3364 | 0.2429 |
| IRT Difficulty (Qwen-only) | 0.2839 | 0.2031 |

Table 14: Correlation analysis of empirical error rate, IRT difficulty and human-rated difficulty. Spearman & Kendall measure the strength and direction of the monotonic relationship (or rank-order correlation) between two variables.

We also conduct ablation studies by fitting IRT models on the original response matrix without augmentation, and on a matrix using only the Qwen-series models. The results show that the difficulty values derived solely from the Qwen series (11 test-takers) correlate poorly with both empirical accuracy and human difficulty, as the limited number of subjects introduces significant bias into the IRT fit. Besides, the original response matrix without augmentation is also less correlated due to data sparsity. This finding validates the effectiveness of our VAE and Sampling data augmentation methods. By learning from the original response matrix and empirical accuracy to generate a large number of distributionally-consistent virtual responses to expand the matrix, our approach enhances the robustness of the IRT fitting process. This prevents parameter estimation bias caused by an insufficient number of subjects.

## D.2 CASE FOR IRT DIFFICULTY

As previously discussed, although both empirical error rate and IRT difficulty can be regarded as values for LLM Difficulty, empirical error rate's drawback is its assumption of equal weighting for all LLMs, leading to a lack of item discrimination. In Figure 11, we present a case study of two questions. Although both questions have the identical empirical accuracy (**37.14%**), their IRT difficulties differ significantly: Question 1 has an IRT difficulty of **0.599**, whereas Question 2 has an IRT difficulty of **0.778**. This discrepancy arises because the profile of models that answer each question correctly is different. Specifically, four high-ability models including Deepseek-R1-Distill-Qwen-1.5B, Gemini-2.5-Flash, GLM-Z1-AIRX, and Qwen3-235B-A22B (see ability values in Table 6) answer Question 1 correctly but fail to answer Question 2. Therefore, even with the same empirical accuracy, Question 1 is more likely to be solved by high-ability LLMs, thus receiving a lower IRT difficulty score. This case validates the advantage of using IRT difficulty over simple empirical error rate, as it successfully distinguishes item difficulty based on the ability of the solvers, not just the raw pass count.

**Question 1:** Given a $4\times 4$ real matrix $T$ such that $T^4=0$, determine which of the following sequences $k_1, \\ k_2,\\ k_3,\\ k_4$ is NOT a possible combination for the nullity of $T^i$:\n\n1. $k_1 = 3,\\ k_2 = 4,\\ k_3 = 4,\\ k_4 = 4$\n2. $k_1 = 1,\\ k_2 = 3,\\ k_3 = 4,\\ k_4 = 4$\n3. $k_1 = 2,\\ k_2 = 4,\\ k_3 = 4,\\ k_4 = 4$\n4. $k_1 = 2,\\ k_2 = 3,\\ k_3 = 4,\\ k_4 = 4$\n\nNote: Since $T^4=0$, $T$ is nilpotent and its eigenvalues are all $0$. Consider the properties of nilpotent matrices to determine the answer.
**Answer:** 2

**Empirical Accuracy:** 37.14%
**IRT Difficulty:** 0.599

**Question 2:** In the right triangle \\(ABC\\), the altitude \\(BH\\) is drawn to the hypotenuse \\(AC\\). Points \\(X\\) and \\(Y\\) are the centers of the circles inscribed in triangles \\(ABH\\) and \\(CBH\\) respectively. The line \\(XY\\) intersects the legs \\(AB\\) and \\(BC\\) at points \\(P\\) and \\(Q\\). Given that \\(BH = h\\), find the area of triangle \\(BPQ\\).
**Answer:** \\dfrac{h^2}{2}

**Empirical Accuracy:** 37.14%
**IRT Difficulty:** 0.778

Figure 11: A case study of two questions with identical empirical accuracy and different IRT difficulties.

## D.3 DISTRIBUTION SHIFT

To validate the stability of our IRT parameter estimations, we collect additional responses for the same $2,000$ items using five extra models: Qwen2-72B, Mistral-large, GPT-oss-20B, Qwen3-1.7B, and GPT-4.1-nano. Among these, Qwen2-72B and Qwen3-1.7B are classified as "lower-performing LLMs" as their accuracies fell below the $67.98\%$ average accuracy of our initial 35 LLMs. We then conduct two experiments: first, we added only these weaker LLMs to the original 35-LLM response matrix; second, we added all 5 new models to the original 35-LLM matrix. We then fit IRT parameters for both augmented matrices and calculate the Spearman and Kendall correlation coefficients against the original IRT difficulty parameters derived from the initial 35 LLMs. The results (shown in Table 15) indicate that in both scenarios—whether adding only the weaker models or all five additional models—the changes in the Spearman and Kendall correlation coefficients are minimal. This demonstrates the stability of our method, confirming that our IRT parameter estimations are robust and not significantly perturbed by the inclusion of a certain amount of new response data.

| vs 35-LLMs IRT Difficulty | Spearman | Kendall |
|---|---|---|
| 35-LLMs + lower-performing 2-LLMs | 1.00 | 0.9968 |
| 35-LLMs + 5-LLMs | 1.00 | 0.9954 |

Table 15: Correlation analysis between the original 35-LLM fitted IRT difficulty and those fitted with 2 additional lower-performing LLMs or all 5 additional LLMs.

