# OpenReview forum: "RIDE: Difficulty Evolving Perturbation with Item Response Theory for Mathematical Reasoning"
_ICLR.cc/2026/Conference — Submitted to ICLR 2026_

### Official Review · Reviewer_GWMD · 2025-10-30

**Soundness:** 2
**Presentation:** 3
**Contribution:** 2
**Rating:** 4
**Confidence:** 3

**Summary:**

This work builds purtabated benchmarks to evaluate the robustness of mathematical reasoning LLMs using the proposed RIDE pipeline. RIDE differs from previous rule-based methods ('rule-based' in the paper means writing explicit rules in the prompt for LLM to synthesize a new question) in that:
(1) Their question rewriting model, RIDE-8B, is trained from Qwen3-8B with an SFT stage that distills from GPT-5 and a followed RL stage using an IRT(Item Response Theory)-based difficulty-ranking model as part of the source of the reward signal during GRPO training.
(2) Their prompt to generate purturbated questions do not explicitly contain any specific rules such as numerical variation. Instead, they only provide some general instructions such as to increase the difficulty, and not to turn the original problem into multiple sub-problems.

The authors show that RIDE is superior to one rule-based counterpart (GSM-Plus prompting strategy) under the quality evaluation by GPT-5. After curation by human annotators/advanced LLMs on the RIDE-generated questions, they build and release RIDE-AIME and RIDE-AMC benchmarks and a training dataset DeepMath-RIDE.

**Strengths:**

Originality: (1) The question rewriting prompt is substantially different from the previous rule-based ones and allows strong LLMs to create more diverse problems than the rule-based counterparts.
(2) In order to mitigate the scarcity of student responses for training the difficulty-ranking model, this work proposes to use VAE/Sampling augmentation strategies to obtain extra IRT rows for better training.

Clarity: The RIDE pipeline is clearly stated and the overall workflow shown in Figure 2 is very easy to understand. Experimental setups are introduced with enough details.

**Weaknesses:**

1. The significance of using RL to train the model is not defended in the paper. It is important for the authors to show that the RL stage indeed improves the model performance to defend their contributions on the difficulty-ranking model used for providing reward signal. For example, adding extra rows in Table 2 to show the result of RIDE without IRT reward and RIDE without RL training stage would be ideal.

2. The significance of building such benchmarks today seems not that clear to me. Since the ceiling of the difficulty of the rewriting question is not controlled, it is expected that the model performance will definitely decrease due to the increase of difficulty. As a result, it is not that easy to claim that the model is not robust simply because its performance degrades on RIDE-AIME/AMC. In contrast, the MATH-P-hard benchmark proposed in Huang et al. 2025 is a pure human-annotated purturbated dataset, so that they can make sure despite that the original solution paths are no longer applicable, the actual 'difficulty' of the question is not increased, which allows for focusing on the robostness of math reasoning only.

3. There is no systematic analysis on how RIDE-8B tends to rewrite the original question. The paper only shows a case study in section 5.3. Adding a statistics on the types of rewriting for a certain quantity of sampled questions could help strenthen the argument.

**Questions:**

My major concerns and questions have been proposed above in the weaknesses. Below are some extra questions on the details:
1. Can you please show the results of RIDE-Qwen3-1.7B and RIDE-Qwen3-4B on AIME-25? It is acceptable that the performance maintains or even degrades, I am just curious about whether the results are consistent.
2. Why in Table 2, for AMC-23, the sum of average win rates of rule-based and RIDE is 99.68% rather than 100.0%?
3. The right panel of Figure 4 shows that the pass@n performance of Qwen2.5-Math-72B hardly increases as n increases from 1 to 8. This is quite out of my expectation since the Qwen2.5-Math technical report shows that maj@64 result has a clear margin over pass@1 result. Could you help explain this?

**Details Of Ethics Concerns:**

The annotator compensation is not clearly stated in the paper. I will appreciate if the authors can provide how much the annotators are paid for an hour.

---

> ### Author Response · Authors · 2025-11-20
> **Author Response to Reviewer GWMD Part 1**
>
> Dear Reviewer GWMD,
>
> We sincerely thank the reviewer for the high-quality feedback and insightful comments. Please allow us to respond to the weaknesses and questions raised. (Part 1)
>
>
>
> > **W1: The significance of using RL to train the model is not defended in the paper. It is important for the authors to show that the RL stage indeed improves the model performance to defend their contributions on the difficulty-ranking model used for providing reward signal. For example, adding extra rows in Table 2 to show the result of RIDE without IRT reward and RIDE without RL training stage would be ideal.**
>
> Following your valuable suggestions, we have added new experiments specifically for the Reinforcement Learning (RL) stage:
>
> 1. **Ablation on Reinforcement Learning.** We first evaluate the output of the model trained only with SFT. We report the win rate of the SFT-only model's rewritten questions against our final rewritten questions.
>
> 2. **Ablation on Difficulty Reward $R_{diff}$.** We then remove the difficulty reward $R_{diff}$ from the RL objective and compare the resulting model's rewritten questions with our final rewritten questions, again reporting the win rate. The results (shown in Table 2 of our revised paper)  are summarized below:
>
>    | **Compared Method**       | **Dataset** | **Baseline (Average Win Rate)** | **RIDE (Average Win Rate)** | **Baseline (Consistent Win Rate)** | **RIDE (Consistent Win Rate)** |
>    | ------------------------- | ----------- | ------------------------------- | --------------------------- | ---------------------------------- | ------------------------------ |
>    | **SFT-only**              | AMC-23      | 38.75                           | **60.00**                   | 33.33                              | **53.33**                      |
>    |                           | AIME-24     | 36.67                           | **62.22**                   | 26.67                              | **52.22**                      |
>    | **w/o $R_{\text{diff}}$** | AMC-23      | 43.75                           | **54.17**                   | 35.83                              | **47.50**                      |
>    |                           | AIME-24     | 27.78                           | **72.22**                   | 21.11                              | **65.56**                      |
>
> Furthermore, we computed the Difficulty Score using the average predicted probability from our difficulty ranker, which indicates the likelihood of the rewritten question being more difficult than the original.  We compared the performance of Qwen3-8B, the SFT-only baseline (RIDE-8B-SFT), and our RIDE-8B model. We also conducted **an ablation study on the Correctness Reward $R_{cor}$.** Specifically, we utilized GPT-5 to judge the Correctness of the questions rewritten by our RIDE-4B model. The results (shown in Table 5 of our revised paper) are presented below:
>
> | **Model**                    | **Difficulty** | **Correctness** |
> | ---------------------------- | -------------- | --------------- |
> | Qwen3-8B                     | 74.61          | 40.00           |
> | RIDE-8B-SFT                  | 78.78          | 40.00           |
> | RIDE-8B                      | **80.18**      | **42.50**       |
> | RIDE-4B                      | -              | **45.00**       |
> | RIDE-4B w/o $R_{\text{cor}}$ | -              | 32.50           |
>
> In summary, our final RL-trained model outputs rewritten questions that are significantly more complete, of higher difficulty, and possess better correctness compared to the SFT-only baseline model.

---

> ### Author Response · Authors · 2025-11-20
> **Author Response to Reviewer GWMD Part 2**
>
> Dear Reviewer GWMD,
>
> We sincerely thank the reviewer for the high-quality feedback and insightful comments. Please allow us to respond to the weaknesses and questions raised. (Part 2)
>
>
>
> > **W2:** **The significance of building such benchmarks today seems not that clear to me. Since the ceiling of the difficulty of the rewriting question is not controlled, it is expected that the model performance will definitely decrease due to the increase of difficulty. As a result, it is not that easy to claim that the model is not robust simply because its performance degrades on RIDE-AIME/AMC. In contrast, the MATH-P-hard benchmark proposed in Huang et al. 2025 is a pure human-annotated purturbated dataset, so that they can make sure despite that the original solution paths are no longer applicable, the actual 'difficulty' of the question is not increased, which allows for focusing on the robostness of math reasoning only.**
>
> Our idea is not to introduce several new benchmarks, but to provide an **automated testing framework** for rewriting mathematical reasoning questions, which perturbs existing benchmarks to test model robustness. As current LLM reasoning capabilities are rapidly improving, former static benchmarks with fixed difficulty levels may soon become ineffective to cause performance drop. Therefore, it is necessary to consider dynamically increasing difficulty to maintain the benchmark’s relevance, enabling **real-time robustness evaluation** for the latest LLMs. Moreover, constructing manually annotated data for MATH-Perturb is highly costly and does not generalize well to additional datasets. Consequently, our method offers significant advantages in terms of real-time adaptability and generalization.

---

> > ### Comment · Reviewer_GWMD · 2025-11-20
> > **Thank you for the careful response**
> >
> > I appreciate the efforts the authors have made to justify the soundness and significance of the rework. I am convinced by most of the response. However, I still have some questions in your response to my concerns on the significance of building RIDE-AIME and RIDE-AMC.
> >
> > >Our idea is not to introduce several new benchmarks, but to provide an automated testing framework for rewriting mathematical reasoning questions, which perturbs existing benchmarks to test model robustness. As current LLM reasoning capabilities are rapidly improving, former static benchmarks with fixed difficulty levels may soon become ineffective to cause performance drop. Therefore, it is necessary to consider dynamically increasing difficulty to maintain the benchmark’s relevance, enabling real-time robustness evaluation for the latest LLMs. Moreover, constructing manually annotated data for MATH-Perturb is highly costly and does not generalize well to additional datasets. Consequently, our method offers significant advantages in terms of real-time adaptability and generalization.
> >
> > 1. I feel that the response is not addressing my concern directly. I apologize for that my initial comment was not lear, let me restate it: My concern is that, when you build a more difficult benchmark, how could you tell that the performance drop comes from the increase of difficulty, or from the lack of robustness of LLM math reasoning?
> >
> > 2. Since you have claimed in the paper that
> > > Using GPT-5, we filter out questions that are unsolvable or have incorrect answers to construct the final perturbed benchmarks RIDE-AMC and RIDE-AIME, which are **further manually corrected to ensure answer accuracy, with annotators holding at least a bachelor’s degree**
> >
> > , I believe this procedure is not full-automated. And the paper does not provide any evidence that the annotation cost is much cheaper than that of building benchmarks like MATH-Perturb.

---

> > > ### Author Response · Authors · 2025-11-23
> > > **Author Response to Reviewer GWMD Part 1**
> > >
> > > Dear Reviewer GWMD,
> > >
> > > We sincerely thank you for the feedback and insightful comments.  Please allow us to respond to your new concern.
> > >
> > > > **1. When you build a more difficult benchmark, how could you tell that the performance drop comes from the increase of difficulty, or from the lack of robustness of LLM math reasoning?**
> > >
> > > We argue that the verification of model robustness and the evolution of question difficulty can **coexist.** We define the robustness of LLMs in mathematical reasoning as follows: for a question $x$ and the model's response correctness $y \in \{0, 1\}$, given a semantic-preserving transformation family $\mathcal{T}$, if $f(x)=y$, then for the vast majority of $t \in \mathcal{T}$, $f(t(x)) = y$ holds. **Figure 9** shows that the semantic similarity between our rewritten and original questions consistently exceeds **80%**, confirming that performance degradation does not stem from semantic disparity. Building on this, we decouple question difficulty from robustness verification.
> > >
> > > - **Case analysis.** As detailed in **Section 5.3 (from line 400 to line 418)**, we identify **"surface-level modifications"** that preserve reasoning depth. An example follows:
> > >
> > > > | **Question**           | **Content**                                                  |
> > > > | ---------------------- | ------------------------------------------------------------ |
> > > > | **Original Question**  | In a table tennis tournament every participant played every other participant exactly once. Although there were twice as many right-handed players as left-handed players, the number of games won by left-handed players was 40% more than the number of games won by right-handed players. (There were no ties and no ambidextrous players.) What is the total number of games played? |
> > > > | **Rewritten Question** | In a round-robin table tennis tournament, every participant plays every other participant exactly once (no ties, no ambidextrous players). Let R be the number of right-handed players and L be the number of left-handed players, **with R = 2L.** Over the whole tournament, the total number of games won by left-handed players is 40% more than the total number of games won by right-handed players. Find the total number of games played. |
> > >
> > > In this example, the evolution of the question is manifested in the formalized mathematical conditions and changes in phrasing, which do not impact the depth of reasoning required, making this a case for verifying model robustness. For example, Qwen2.5-72B answered the original question correctly but failed on the rewritten question.
> > >
> > > - **Quantitative analysis 1: difficulty-stratified performance.** We collected similar surface-only rewrites and categorized them into three difficulty levels—easy, medium, and hard—based on the accuracy rates of 10 LLMs on the original questions. The questions rewritten using the surface-only strategy maintain a **reasoning depth consistent** with the original questions, leaving the surface form as the sole variable. If the model's performance degradation were solely dependent on difficulty, the magnitude of the performance drop on the rewritten questions should vary significantly across the three difficulty levels. Conversely, if question difficulty and robustness verification can be **decoupled**, the variations in performance decline across the three difficulty levels should be insignificant. We evaluated the performance degradation between the original and rewritten questions across different difficulty levels:
> > >
> > > | **Level**  | **Original Accuracy** | **Rewritten Accuracy** | Performance Drop |
> > > | ---------- | --------------------- | ---------------------- | ---------------- |
> > > | **Easy**   | 80.00                 | 68.57                  | 11.43            |
> > > | **Medium** | 68.57                 | 55.71                  | 12.86            |
> > > | **Hard**   | 62.50                 | 43.75                  | 18.75            |
> > >
> > > The results indicate that surface-level rewriting induces performance degradation even on easy questions, thereby highlighting the necessity of robustness verification. Moreover, the difference of the performance drop across difficulty levels is not quite significant, with a particularly negligible variation in the magnitude of change between **easy and medium** questions.

---

> > > ### Author Response · Authors · 2025-11-23
> > > **Author Response to Reviewer GWMD Part 2**
> > >
> > > - **Quantitative analysis 2: McNemar’s test and model performance analysis.** We additionally recorded the counts of "Correct on Original, Incorrect on Rewritten" and "Incorrect on Original, Correct on Rewritten" for each paired sample in the surface-level rewriting dataset and calculated the p-value using **McNemar's test.** The resulting **p-value** of **$2.3847 \times 10^{-11}$** is substantially **lower than 0.05**, revealing a significant asymmetry. This demonstrates that the performance decline originates from the model's non-robustness rather than changes in difficulty. Furthermore, we evaluated the accuracy of two LLMs across different types of rewriting:
> > >
> > > | Type             | Qwen2.5-Math-72B | Llama4-Scout |
> > > | :--------------- | :--------------- | :----------- |
> > > | **Surface-only** | 18.2             | 13.6         |
> > > | **Mixed**        | 48.4             | 17.6         |
> > > | **Deep-only**    | 47.1             | 41.9         |
> > >
> > > Results show that surface-level rewrites yielded the lowest accuracy, even below deep rewrites, confirming our method's efficacy for robustness verification.
> > >
> > > In summary, through case study and statistical analysis, we decoupled question difficulty from robustness verification and found that the two are **orthogonal**, allowing them to **coexist within our method**.
> > >
> > > > **2. I believe this procedure is not full-automated. And the paper does not provide any evidence that the annotation cost is much cheaper than that of building benchmarks like MATH-Perturb.**
> > >
> > > We acknowledge that the description in our initial submission was not sufficiently precise. In reality, model-based filtering is fully capable of eliminating erroneous data. The human review was employed solely as a safeguard to ensure the released benchmark is absolutely error-free. We did not perform any manual modifications to the final data, but merely conducted verification checks. We have **modified the description** in the revised paper.
> > >
> > > We validated the capability of LLMs to serve as verifiers/filters. Specifically, we input the math question, a model-generated solution and answer into the verifying LLM, instructing it to determine the correctness of the solution. The classification metrics for GPT-5-mini are as follows:
> > >
> > > | **Response Source** | **Verifier** | **Precision** | **Recall** | **F$_1$** | **Accuracy** |
> > > | ------------------- | ------------ | ------------- | ---------- | --------- | ------------ |
> > > | Qwen2.5-Math-7B     | GPT-5-mini   | 97.96         | 79.17      | 87.57     | 85.70        |
> > > | Gemma3-12B          | GPT-5-mini   | 98.38         | 79.71      | 88.06     | 86.00        |
> > >
> > > The results demonstrate that GPT-5-mini achieves a precision of approximately **98%** (with GPT-5 expected to be even higher), highlighting its effectiveness in filtering out incorrect data. This aligns with the intuition that verifying the correctness of a solution is inherently easier for a model than solving the question from scratch. We can utilize RIDE-8B to generate $n$ sampled outputs for a given question and subsequently filter them to obtain the final result (After the model generates a rewrite question, a filter judges it. If it is correct, the rewrite question is kept; otherwise, a new problem is generated). The manual review was strictly a safeguard against extremely rare false positive cases. In practice, our manual verification confirmed that **model-based filtering is highly reliable** and **requires no manual intervention for modification.** This step was intended solely to guarantee the **absolute correctness** of the benchmark we are releasing. For the purpose of data augmentation in **training datasets**, model-based filtering is sufficient to handle the entire workflow **without any human involvement.**
> > >
> > > In contrast, Math-Perturb relies on manual annotation, requiring extensive human modification of question content. The approach lacks scalability and cannot be easily transferred to new datasets. Conversely, our method can be directly applied to perform inference on other datasets, demonstrating significantly **stronger generalization capabilities.**

---

> > > ### Author Response · Authors · 2025-11-26
> > >
> > > Dear Reviewer GWMD,
> > >
> > > I hope this message finds you well. As the discussion period is nearing its end with less than one week, I wanted to ensure that we have addressed all your concerns satisfactorily. If there are any additional points or feedback you'd like us to consider, please let us know. Your insights are invaluable to us, and we're eager to address any remaining issues to improve our work.
> > >
> > > Thank you for your time and effort in reviewing our paper.

---

> ### Author Response · Authors · 2025-11-20
> **Author Response to Reviewer GWMD Part 3**
>
> Dear Reviewer GWMD,
>
> We sincerely thank the reviewer for the high-quality feedback and insightful comments. Please allow us to respond to the weaknesses and questions raised. (Part 3)
>
>
> > **W3:** **There is no systematic analysis** **on** **how RIDE-8B tends to** **rewrite** **the original question. The paper only shows a case study in section 5.3. Adding** **a** **statistics on the types of rewriting for a certain quantity of sampled questions could help** **strenthen** **the argument.**
>
> To systematically analyze these rewrites, we distill 6 most common rewrite patterns from the questions in the RIDE-AIME and RIDE-AMC datasets: (1) Wording and Paraphrasing Modification, (2) Distracting Info, (3) Numerical Substitution, (4) Add Extra Steps, (5) Change Constraints, and (6) Change Target. (Detailed descriptions are provided in Appendix B.4). First, we counted the frequency with which each rewriting strategy appeared in our set of rewritten questions:
>
> | **Type**                                  | **Frequency (%)** |
> | ----------------------------------------- | ----------------- |
> | **Wording and Paraphrasing Modification** | 65.7              |
> | **Insertion of Distracting Information**  | 11.4              |
> | **Numerical Value Substitution**          | 17.1              |
> | **Addition of Extra Variables/Steps**     | 32.9              |
> | **Change of Constraints or Rules**        | 44.3              |
> | **Change of the Target Expression**       | 28.6              |
>
> Secondly, we tallied the distribution of the number of rewriting strategies present in the rewritten questions:
>
> | **Number of strategies** | **Percentage** |
> | ------------------------ | -------------- |
> | 1                        | 35.7           |
> | 2                        | 35.7           |
> | 3                        | 21.4           |
> | 4                        | 7.1            |
>
> Over 60% of the rewritten questions are composed of two or more strategies. To further investigate the impact on reasoning depth, we categorize strategies 1-3 as Surface Strategies, which have a minor impact on reasoning depth, and strategies 4-6 as Deep Strategies, which significantly affect the required reasoning.  We categorize the questions based on the type of strategies used: those employing only Surface Strategies, those employing only Deep Strategies, and those employing a Mixture of both. The frequency statistics are as follows：
>
> | **Type**         | **Frequency (%)** |
> | ---------------- | ----------------- |
> | **Surface-only** | 31.4              |
> | **Mixed**        | 44.3              |
> | **Deep-only**    | 24.3              |
>
> Statistics show that rewrites using a mix of both Surface and Deep strategies are the most common (44.3%). Finally, we evaluate the accuracy of Qwen2.5-Math-72B and Llama4-Scout on questions corresponding to each strategy：
>
> | **Strategy Type**                         | Qwen2.5-Math-72B | Llama-4-Scout |
> | ----------------------------------------- | ---------------- | ------------- |
> | **Wording and Paraphrasing Modification** | 32.6             | 28.3          |
> | **Insertion of Distracting Information**  | 75.0             | 62.5          |
> | **Numerical Value Substitution**          | 41.7             | 25.0          |
> | **Addition of Extra Variables/Steps**     | 56.5             | 47.8          |
> | **Change of Constraints or Rules**        | 48.4             | 41.9          |
> | **Change of the Target Expression**       | 45.0             | 35.0          |
>
>  We find that performance varies across different types.  Detailed Analysis and visualization result are shown in Section 5.3 of our revised paper.
>
>
>
> > **Q1: Can you please show the results of RIDE-Qwen3-1.7B and RIDE-Qwen3-4B on AIME-25? It is acceptable that the performance maintains or even degrades, I am just curious about whether the results are consistent.**
>
> For the AIME25 dataset, our RIDE-Qwen-1.7B achieves a Pass@1 result of **13.33**, compared to 6.67 for the Qwen3-1.7B baseline. However, for the 4B model size, our performance remains consistent with the baseline. This discrepancy is likely due to the fact that we utilized only a subset of the available data for training. We plan to address this by continuing our training process using the full dataset in subsequent attempts to further improve the performance.
>
>
>
> > **Q2: Why in Table 2, for AMC-23, the sum of average win rates of rule-based and RIDE is 99.68% rather than 100.0%?**
>
> As we allow the evaluator model to output a **"tie"** result, these instances are included in the denominator total comparisons but excluded from the numerator (wins) when calculating the average win rate and consistent win rate.  We have already emphasized this in the revised paper.

---

> ### Author Response · Authors · 2025-11-20
> **Author Response to Reviewer GWMD Part 4**
>
> Dear Reviewer GWMD,
>
> We sincerely thank the reviewer for the high-quality feedback and insightful comments. Please allow us to respond to the weaknesses and questions raised. (Part 4)
>
> > **Q3: The right panel of Figure 4 shows that the pass@n performance of Qwen2.5-Math-72B hardly increases as n increases from 1 to 8. This is quite** **out of** **my expectation since the Qwen2.5-Math** **technical report** **shows that maj@64 result has a clear margin over pass@1 result. Could you help explain this?**
>
> In Qwen2.5-Math Technical Report, the results of Pass@1 and Maj@64 are shown as:
>
> | **Model**                   | AIME--24               | AMC-23                  |
> | --------------------------- | ---------------------- | ----------------------- |
> | **Qwen2-Math-72B-Instruct** | 6/30                   | 24/40                   |
> |                             | 8/30$_{\text{maj@64}}$ | 29/40$_{\text{maj@64}}$ |
>
> In our evaluation of Qwen2.5-Math-72B, we observed a Pass@1 result of **26.67% (8/30)** on the AIME24 dataset, which is higher than the 6/30 reported by the technical report and consistent with their reported maj@64 result. Our Pass@1 result on the AMC-23 dataset is **70% (28/40)**, also exceeding their reported figure. This discrepancy might stem from differences in the inference temperature setting, as the original paper did not specify the parameters used during inference. We further tested the Pass@64 result and achieved **75% (30/40),** which is also higher than their reported maj@64 result. This is expected, as Pass@n is generally considered a more relaxed metric compared to majority voting. Also, recent work has shown that AIME-24 is heavily contaminated for many state-of-the-art LLMs[1], and thus is no longer a reliable standalone indicator of general mathematical reasoning ability. Consequently, it is likely that Qwen2.5-Math is heavily fitted to existing math questions, which further corroborates the necessity of our method.
>
> [1] MathArena: Evaluating LLMs on Uncontaminated Math Competitions
>
> We have marked the added content in **blue** in the revised paper. Thank you again for your careful reading and valuable review.

---

### Official Review · Reviewer_iD67 · 2025-10-30

**Soundness:** 3
**Presentation:** 4
**Contribution:** 3
**Rating:** 6
**Confidence:** 4

**Summary:**

The paper introduces RIDE, a novel adversarial benchmark framework for evaluating the mathematical reasoning robustness of large language models (LLMs). Traditional rule-based perturbation methods for benchmark augmentation often yield ill-posed or trivial question variants. To address this, RIDE integrates Item Response Theory (IRT) into a reinforcement learning–based question rewriting pipeline to obtain a rewriter as well as rewritten datasets that has considerable hardness.

**Strengths:**

1. The paper is well-detailed in its presentation about the methodology and experiments. A comprehensive pipeline is designed for data augmentation that includes reward model training and rewriter training.
2. The paper introduced a simple model (IRT) that handsomely credits the hardness of the problem generates.

**Weaknesses:**

1. Given the limited LLM resources, the paper adopted multiple augmentation methods to extend problem-response data from LLM responses, including a VAE/sampling augmentation and the training of a pairwise difficulty ranker. However, as both methods act as a bootstrapping of existing responses, it is unclear that whether this approach exacerbates the overfitting of reward modeling.

It is also unclear to the reader why it is necessary to augment the response matrix using VAE method or sampling method. The paper did not effectively explain why these augmentation methods are useful and did not incorporates other baselines as reward models.

2. A major novelty of the paper is to introduce a rewritter model that automatically rewrites problems. However, it still requires human-in-the-loop to produce the final augmented datasets. This weakens the need for training a rewritter. However, the paper provides no discussion on the edges of the current methodology over previous methodologies that incorporates human rewritting the problems. (e.g. https://arxiv.org/abs/2502.06453)

3. In the experiments, many SOTA LLMs does not display significant performance drop for the more difficult dataset, and there is a lack of discussion. The definition of PDR also tends exaggerate the actual effect on weaker models.

**Questions:**

1. Regarding the first point in the weakness, can you elaborate why we need to augment the response matrix, and especially why sampling can help the training of the reward model?

2. Regarding the second point in the weakness, can you elaborate why the method is more efficient than human rewritting?

---

> ### Author Response · Authors · 2025-11-20
> **Author Response to Reviewer iD67 Part 1**
>
> Dear Reviewer iD67,
>
> We sincerely thank the reviewer for the high-quality feedback and insightful comments. Please allow us to respond to the weaknesses and questions raised. (Part 1)
>
> > **W1: Given the limited** **LLM** **resources, the paper adopted multiple augmentation methods to extend problem-response data from LLM responses, including a VAE/sampling augmentation and the training of a pairwise difficulty ranker. However, as both methods act as a bootstrapping of existing responses, it is unclear that whether this approach exacerbates the** **overfitting** **of reward modeling.** **It is also unclear to the reader why it is necessary to augment the response matrix using** **VAE** **method or sampling method. The paper did not effectively explain why these augmentation methods are useful and did not incorporates other baselines as reward models.**
> >
> > **Q1: Regarding the first point in the weakness, can you elaborate why we need to augment the response matrix, and especially why sampling can help the training of the reward model?**
>
> Table 4 and Table 10 demonstrate that our two data augmentation methods improve the AUC-ROC values for missing response prediction, thereby demonstrating that our approach enhances generalization rather than exacerbating overfitting. Furthermore, we employed data augmentation to enhance the stability of the IRT prediction results. We conducted IRT modeling based on the response matrix with augmented data, the original response matrix, and the response matrix containing only Qwen series models, respectively. We then compared the resulting difficulty distributions with AoPS (human difficulty) using Spearman and Kendall correlation coefficients. The results are as follows:
>
> | **vs Human-Rated Difficulty**   | **Spearman** | **Kendall** |
> | ------------------------------- | ------------ | ----------- |
> | IRT Difficulty                  | 0.3418       | 0.2468      |
> | IRT Difficulty w/o augmentation | 0.3364       | 0.2429      |
> | IRT Difficulty (Qwen-only)      | 0.2839       | 0.2031      |
>
> Incorporating data augmentation results in higher consistency between our IRT difficulty and human-rated difficulty rankings. In contrast, relying solely on the Qwen series leads to instability due to data scarcity, whereas our data augmentation method effectively **enhances this stability and robustness.**
>
> Besides, we believe the Sampling method incorporates a supervision signal based on empirical accuracy, which acts as a safeguard against interference from anomalous test takers. For example, if an anomalous test taker answers hard questions correctly but misses easy ones, this would interfere with both VAE augmentation and IRT convergence. Utilizing empirical accuracy effectively suppresses this noise, ensuring that the IRT difficulty results align with empirical data, thereby improving overall robustness. We will provide a more detailed elaboration in the revised paper.
>
>
>
> > **W2: A major novelty of the paper is to introduce a rewritter model that automatically rewrites problems. However, it still requires** **human-in-the-loop** **to produce the final augmented datasets. This weakens the need for training a** **rewritter. However, the paper provides no discussion on the edges of the current methodology over previous methodologies that incorporates human rewritting the problems.**
> >
> > **Q2: Regarding the second point in the weakness, can you elaborate why the method is more efficient than human rewriting?**
>
> 1. Please allow me to clarify that our framework **does not require human intervention.** Our manual verification was solely intended to ensure that the model-filtered benchmark is completely correct (as we plan to open-source these data for future research, absolute certainty is essential). In fact, the results of the manual check confirm that model filtering is entirely feasible, and human effort made no actual modifications to the overall process.
> 2. Furthermore, given the rapid iteration and update cycles of current datasets and benchmarks, we aimed to use an **automated** approach for benchmark perturbation to detect data leakage and pattern matching issues in models. In contrast, the Math-Perturb relies on 12 PhD students to manually rewrite large-scale data, leading to excessive manpower costs and making automation difficult to implement. In contrast, our method limits human involvement solely to the final verification of the open-source benchmark, with no manual content modification required. The entire pipeline is fully automated by models, ensuring greater efficiency and generalizability. We will provide a more detailed elaboration in the revised paper.

---

> ### Author Response · Authors · 2025-11-20
> **Author Response to Reviewer iD67 Part 2**
>
> Dear Reviewer iD67,
>
> We sincerely thank the reviewer for the high-quality feedback and insightful comments. Please allow us to respond to the weaknesses and questions raised. (Part 2)
>
>
>
> > **W3: In the experiments, many SOTA LLMs does not display significant performance drop for the more difficult dataset, and there is a lack of discussion. The definition of PDR also tends exaggerate the actual effect on weaker models.**
>
> Significant performance degradation was also observed in certain proprietary models, such as **Claude-4-Opus (from 53.33% to 36.67%),** on our rewritten benchmark. In contrast, the decline was negligible in some of the sota LLMs, validating their superior robustness, which aligns with our initial motivation to verify data leakage and assess robustness against pattern matching.
>
> For PDR, while we acknowledge that PDR, as a ratio metric, can be sensitive to low baselines, we maintain that it validly measures the "stability of acquired knowledge". if a weak model loses its few correct answers after perturbation, it rightly exposes a reliance on superficial pattern matching rather than genuine reasoning. Empirically, our results in Table 1 demonstrate that low baseline accuracy does not inherently lead to inflated PDR. For instance, **Qwen2.5-7B** has a low baseline accuracy (10.0%) on AIME-24 yet achieves a 0.00% PDR, proving that the metric effectively distinguishes true robustness from numerical artifacts. To ensure a comprehensive evaluation, we explicitly also report absolute Pass@1 scores alongside PDR.
>
>
>
> We have marked the added content in **blue** in the revised paper. Thank you again for your careful reading and valuable review.

---

> ### Author Response · Authors · 2025-11-26
>
> Dear Reviewer iD67,
>
> I hope this message finds you well. As the discussion period is nearing its end with less than one week, I wanted to ensure that we have addressed all your concerns satisfactorily. If there are any additional points or feedback you'd like us to consider, please let us know. Your insights are invaluable to us, and we're eager to address any remaining issues to improve our work.
>
> Thank you for your time and effort in reviewing our paper.

---

### Official Review · Reviewer_be6K · 2025-11-01

**Soundness:** 3
**Presentation:** 4
**Contribution:** 3
**Rating:** 4
**Confidence:** 4

**Summary:**

This paper introduces RIDE, a novel framework for generating adversarially perturbed mathematical reasoning questions to robustly evaluate LLMs. The core problem addressed is that high performance on existing benchmarks may not reflect true reasoning ability due to data leakage or superficial pattern matching. RIDE's main contribution is a principled approach to evolving question difficulty. It uses Item Response Theory (IRT) to model difficulty, leveraging a cohort of 35 LLMs as "students" to generate a response matrix. The IRT-derived difficulty scores are then used to train a pairwise difficulty ranker. This ranker provides a reward signal for a RL agent, which is trained to rewrite existing math problems to be more challenging. The authors demonstrate that the resulting benchmarks (RIDE-AMC, RIDE-AIME) cause a significant performance drop across 26 powerful LLMs, exposing their brittleness. The paper also shows the utility of the generated data for augmenting training sets.

**Strengths:**

1.  The paper's primary strength is its innovative use of Item Response Theory to formalize and quantify the concept of "question difficulty." The RIDE framework provides a more systematic and data-driven way to evolve difficulty.

2.  The overall technical pipeline is well-conceived and executed. Key design choices are commendable.

3.  The authors test a wide range of 26 state-of-the-art proprietary and open-source models, providing strong evidence for their claims.

4.  The paper is exceptionally well-written and easy to follow. The figures are particularly effective.

**Weaknesses:**

1.  The entire framework hinges on the IRT difficulty estimates, which are derived from the performance of a specific cohort of 35 LLMs. This raises a question: does the framework measure intrinsic mathematical difficulty, or does it measure "difficulty-for-LLMs"? The generated questions might be overfitting to exploit common failure modes of the current llms rather than becoming more difficult in a way that would also challenge a human.

2. The RL training process relies heavily on GPT-5-mini for the correctness reward and GPT-5 for filtering and evaluation. This creates a potential circular dependency and a performance ceiling. The rewriting model (RIDE-8B) might simply be learning to generate problems that are difficult for its peers but still solvable by its "teacher." This undermines the claim of creating truly adversarial examples and instead frames it as a distillation process from a much stronger, proprietary model.

3.  While the RL approach allows for more freedom than rule-based methods, the rewrites might still converge to a limited set of "tricks" to fool LLMs (e.g., adding distractors, increasing numerical complexity, adding specific constraints).

**Questions:**

1.  Could you elaborate on the potential bias introduced by the specific choice of 35 LLMs for the student pool? How might the IRT difficulty parameters change if the pool included more n weaker models?

2.  How does the framework handle cases where the teacher model (GPT-5-mini) provides an incorrect correctness reward? Have you quantified the error rate of the teacher model on the rewritten questions, and how might this noise affect the RL training process?

3.  Could you provide a more qualitative analysis of the rewriting strategies learned by RIDE-8B? Does it learn a diverse set of transformations, or does it tend to rely on a few specific patterns to increase difficulty? For example, what percentage of rewrites involve changing numerical values vs. adding new conceptual constraints vs. altering the problem structure?

---

> ### Author Response · Authors · 2025-11-20
> **Author Response to Reviewer be6k Part 1**
>
> Dear Reviewer be6K,
>
> We sincerely thank the reviewer for the high-quality feedback and insightful comments. Please allow us to respond to the weaknesses and questions raised. (Part 1)
>
>
> > **W1:The entire framework hinges on the IRT difficulty estimates, which are derived from the performance of a specific cohort of 35 LLMs. This raises a question: does the framework measure intrinsic mathematical difficulty, or does it measure "difficulty-for-LLMs"? The generated questions might be overfitting to exploit common failure modes of the current llms rather than becoming more difficult in a way that would also challenge a human.**
>
> Please allow us to clarify that our idea is to propose a framework for perturbing mathematical questions, specifically targeting the issues of pattern matching and data leakage in existing LLMs when performing mathematical reasoning. Our work primarily focuses on this statistically robust **"LLM difficulty."**
>
> The difficulty of math questions for humans is typically measured by reasoning depth, often using standards like AoPS (Art of Problem Solving). Although the DeepMath dataset uses AoPS levels, we found a discrepancy between this human-centric standard and actual LLM performance. We analyzed the mean empirical accuracy of 35 LLMs on 2,000 questions grouped by AoPS levels and found only a moderate negative correlation (Spearman $\rho = -0.5152$). This indicates that the AoPS standard is not strongly equivalent to LLM error rates. We compare the correlation of both empirical error rate (ungrouped) and fitted IRT difficulty against the human-perceived difficulty, again using Spearman and Kendall coefficients:
>
> | **vs Human-Rated Difficulty** | **Spearman** | **Kendall** |
> | ----------------------------- | ------------ | ----------- |
> | Empirical Error Rate          | 0.3362       | 0.2470      |
> | IRT Difficulty                | 0.3418       | 0.2468      |
>
> The results show that both the empirical error rate and our IRT difficulty (fitted on 35 model responses) have a similar correlation with the given human difficulty. Our work is primarily focused on this "LLM difficulty" rather than human-perceived difficulty.
>
> In summary, our framework is designed to generate LLM-effective questions that expose the limitations of LLMs, thereby driving model development. While we acknowledge that it measures 'LLM difficulty,' we have demonstrated that it is not strongly correlated with but is independently valid from the traditional 'human difficulty.' Of course, we also believe that some of the perturbation strategies, such as adding constraints and introducing extra variables, align with the increase in difficulty observed in **human cognition.**
>
>
>
> > **W2: The RL training process relies heavily on GPT-5-mini for the correctness reward and GPT-5 for filtering and evaluation. This creates a potential circular dependency and a performance ceiling. The rewriting model (RIDE-8B) might simply be learning to generate problems that are difficult for its peers but still solvable by its "teacher." This undermines the claim of creating truly adversarial examples and instead frames it as a distillation process from a much stronger, proprietary model.**
>
> GPT-5 model is not involved in the main pipeline but serves purely as a filter. While GPT-5-mini provides a correctness reward signal, the influence of the difficulty reward signal cannot be disregarded. This difficulty reward signal can be traced back to 35 Large Language Models (LLMs) of varying scales and series. Furthermore, our main results (Table 1) confirm that our method causes a performance degradation not only for models of the same scale but also for models with larger parameters and proprietary models. Additionally, we employ the open-source model DeepSeek-V3.2 as a verifier to provide correctness rewards for training **RIDE-DeepSeek**. To assess the impact of this reward, we evaluate the correctness of the generated rewrites against a baseline model trained without the correctness signal:
>
> | Model                   | Correctness |
> | :---------------------- | :---------: |
> | RIDE-GPT-5-mini         |  **45.00**  |
> | RIDE-DeepSeek           |  **37.50**  |
> | RIDE w/o $R_\text{cor}$ |    32.50    |
>
> Results show that while DeepSeek-V3.2 yields lower correctness than GPT-5-mini, it still offers a valid reward signal, resulting in higher correctness than the baseline without any reward. Consequently, we can effectively utilize other models as verifiers, moving beyond a strict dependence on proprietary models such as GPT-5.

---

> ### Author Response · Authors · 2025-11-20
> **Author Response to Reviewer be6k Part 2**
>
> Dear Reviewer be6K,
>
> We sincerely thank the reviewer for the high-quality feedback and insightful comments. Please allow us to respond to the weaknesses and questions raised. (Part 2)
>
>
>
> > **W3: While the** **RL** **approach allows for more freedom than rule-based methods, the rewrites might still converge to a limited set of "tricks" to fool** **LLMs** **(e.g., adding distractors, increasing numerical complexity, adding specific constraints).**
>
> While preserving the core testing points of a question, our rewritten questions formally consist of patterns such as adding conditions and numerical substitution. However, it is precisely the flexible combination of these patterns that increases question difficulty, reduces the sample space available to the LLM, and changes original reasoning paths. This is ultimately reflected in the results as a decline in the model's accuracy. For example,  we distill 6 most common rewrite patterns from the questions in the RIDE-AIME and RIDE-AMC datasets: (1) Wording and Paraphrasing Modification, (2) Distracting Info, (3) Numerical Substitution, (4) Add Extra Steps, (5) Change Constraints, and (6) Change Target. (Detailed descriptions are provided in Appendix B.4). We tallied the distribution of the number of rewriting strategies present in the rewritten questions:
>
> | **Number of strategies** | **Percentage** |
> | ------------------------ | -------------- |
> | 1                        | 35.7           |
> | 2                        | 35.7           |
> | 3                        | 21.4           |
> | 4                        | 7.1            |
>
> Over 60% of the rewritten questions are composed of two or more strategies. To further investigate the impact on reasoning depth, we categorize strategies 1-3 as Surface Strategies, which have a minor impact on reasoning depth, and strategies 4-6 as Deep Strategies, which significantly affect the required reasoning.  We categorize the questions based on the type of strategies used: those employing only Surface Strategies, those employing only Deep Strategies, and those employing a Mixture of both. The frequency statistics are as follows：
>
> | **Type**         | **Frequency (%)** |
> | ---------------- | ----------------- |
> | **Surface-only** | 31.4              |
> | **Mixed**        | 44.3              |
> | **Deep-only**    | 24.3              |
>
> Statistics show that rewrites using a mix of both Surface and Deep strategies are the most common (44.3%).  These results show that our rewritten question is a flexible combination of different strategies, which is also measured by our IRT difficulty reward.
>
>
>
> > **Q1: Could you elaborate on the potential bias introduced by the specific choice of 35** **LLMs** **for the student pool? How might the IRT difficulty parameters change if the pool included more n weaker models?**
>
> To validate the stability of our IRT parameter estimations, we collect additional responses for the same 2, 000 items using five extra models: Qwen2-72B, Mistral-large, GPT-oss-20B, Qwen3-1.7B, and GPT-4.1-nano. Among these, Qwen2-72B and Qwen3-1.7B are classified as ”lower-performing LLMs” as their accuracies fell below the 67.98% average accuracy of our initial 35 LLMs. We then conduct two experiments: first, we added only these weaker LLMs to the original 35-LLM response matrix; second, we added all 5 new models to the original 35-LLM matrix. We then fit IRT parameters for both augmented matrices and calculate the Spearman and Kendall correlation coefficients against the original IRT difficulty parameters derived from the initial 35 LLMs:
>
> | **vs 35-LLMs IRT Difficulty**     | **Spearman** | **Kendall** |
> | --------------------------------- | ------------ | ----------- |
> | 35-LLMs + lower-performing 2-LLMs | 1.00         | 0.9968      |
> | 35-LLMs + 5-LLMs                  | 1.00         | 0.9954      |
>
> The results indicate that in both scenarios—whether adding only the weaker models or all five additional models—the changes in the Spearman and Kendall correlation coefficients are minimal. This demonstrates the stability of our method, confirming that our IRT parameter estimations with augmented response matrix are robust and not significantly perturbed by the inclusion of a certain amounts of new response data.

---

> ### Author Response · Authors · 2025-11-20
> **Author Response to Reviewer be6k Part 3**
>
> Dear Reviewer be6K,
>
> We sincerely thank the reviewer for the high-quality feedback and insightful comments. Please allow us to respond to the weaknesses and questions raised. (Part 3)
>
>
> > **Q2: How does the framework handle cases where the teacher model (GPT-5-mini) provides an incorrect correctness reward? Have you quantified the error rate of the teacher model on the rewritten questions, and how might this noise affect the RL training process?**
>
> We conducted an experiment to evaluate whether the correctness reward model, GPT-5-mini, could accurately judge the correctness given a mathematical question(without standard answers), its generated solution, and its final answer. We provided 2,000 math questions , along with the solutions and answers generated by two test models: Qwen2.5-Math-7B and Gemma3-12B. The results are as follows:
>
> | **Response Source** | **Verifier** | **Precision** | **Recall** | **F$_1$** | **Accuracy** |
> | ------------------- | ------------ | ------------- | ---------- | --------- | ------------ |
> | Qwen2.5-Math-7B     | GPT-5-mini   | 97.96         | 79.17      | 87.57     | 85.70        |
> | Gemma3-12B          | GPT-5-mini   | 98.38         | 79.71      | 88.06     | 86.00        |
>
> The verification accuracy of the GPT-5-mini reached over **85%** and achieved precision of over **97%**. This indicates that  GPT-5-mini can precisely assign low reward signals to incorrect solutions and is effective for filtering erroneous data. The relatively low recall, however, implies that the verifiers tend to be conservative, flagging an otherwise correct solution and answer as incorrect. This may be because the model’s solution contains minor errors even when the final answer is correct. This situation is acceptable, given that rule-based or model-based rewards in Reinforcement Learning with Verified Rewards (RLVR) also exhibit a high number of false negatives [1].  While this situation affects training efficiency, it does not directly lead to training failure, as our verifier assigns low reward signals to incorrect rewrites.
>
> [1] Zhangchen Xu, Yuetai Li, Fengqing Jiang, Bhaskar Ramasubramanian, Luyao Niu, Bill Yuchen Lin, and Radha Poovendran. Tinyv: Reducing false negatives in verification improves rl for llm reasoning, 2025.
>
>
>
> > **Q3: Could you provide a more qualitative analysis of the rewriting strategies learned by RIDE-8B? Does it learn a diverse set of transformations, or does it tend to rely on a few specific patterns to increase difficulty? For example, what percentage of rewrites involve changing numerical values vs. adding new conceptual constraints vs. altering the problem structure?**
>
> Please refer to our response about W3. To systematically analyze these rewrites, we distill 6 most common rewrite patterns  (1) Wording and Paraphrasing Modification, (2) Distracting Info, (3) Numerical Substitution, (4) Add Extra Steps, (5) Change Constraints, and (6) Change Target. (Detailed descriptions are provided in Appendix B.4). First, we counted the frequency with which each rewriting strategy appeared in our set of rewritten questions:
>
> | **Type**                                  | **Frequency (%)** |
> | ----------------------------------------- | ----------------- |
> | **Wording and Paraphrasing Modification** | 65.7              |
> | **Insertion of Distracting Information**  | 11.4              |
> | **Numerical Value Substitution**          | 17.1              |
> | **Addition of Extra Variables/Steps**     | 32.9              |
> | **Change of Constraints or Rules**        | 44.3              |
> | **Change of the Target Expression**       | 28.6              |
>
> Then we tallied the distribution of the number of rewriting strategies present in the rewritten questions and categorized the questions based on the type of strategies used. The results are shown in our response about Weakness 3.
>
> Besides, we evaluate the accuracy of Qwen2.5-Math-72B and Llama4-Scout on questions corresponding to each strategy：
>
> | **Strategy Type**                         | Qwen2.5-Math-72B | Llama-4-Scout |
> | ----------------------------------------- | ---------------- | ------------- |
> | **Wording and Paraphrasing Modification** | 32.6             | 28.3          |
> | **Insertion of Distracting Information**  | 75.0             | 62.5          |
> | **Numerical Value Substitution**          | 41.7             | 25.0          |
> | **Addition of Extra Variables/Steps**     | 56.5             | 47.8          |
> | **Change of Constraints or Rules**        | 48.4             | 41.9          |
> | **Change of the Target Expression**       | 45.0             | 35.0          |
>
>  We find that performance varies across different types.   Detailed Analysis and visualization result are shown in Section 5.3 of our revised paper.
>
> We have marked the added content in **blue** in the revised paper. Thank you again for your careful reading and valuable review.

---

> ### Author Response · Authors · 2025-11-26
>
> Dear Reviewer be6K,
>
> I hope this message finds you well. As the discussion period is nearing its end with less than one week, I wanted to ensure that we have addressed all your concerns satisfactorily. If there are any additional points or feedback you'd like us to consider, please let us know. Your insights are invaluable to us, and we're eager to address any remaining issues to improve our work.
>
> Thank you for your time and effort in reviewing our paper.

---

### Official Review · Reviewer_zuaw · 2025-11-02

**Soundness:** 3
**Presentation:** 3
**Contribution:** 3
**Rating:** 6
**Confidence:** 3

**Summary:**

This paper proposes RIDE, a benchmark-generation and data-augmentation framework that rewrites mathematical reasoning questions to make them more challenging and diagnostically useful. RIDE leverages Item Response Theory (IRT) to estimate intrinsic question difficulty from LLM responses, builds a pairwise difficulty ranker, and trains a reinforcement-learning–based rewriting model that increases question difficulty while maintaining solvability and answer consistency.

**Strengths:**

1. The methodology is rigorous and well-motivated. Difficulty estimation is carefully implemented via variational inference on the Rasch model with data augmentation (VAE + sampling) to stabilize parameter estimation.
2. The pairwise ranker formulation mitigates regression instability in symbolic domains and yields interpretable difficulty scores.
Experimental coverage is extensive: 23 LLMs spanning 0.6 B–1 T parameters, multiple families (Qwen, LLaMA, DeepSeek, GPT, Gemini, etc.), and both open- and closed-source settings.
3. Empirical results are compelling: nearly all models experience significant performance degradation on RIDE-AIME/AMC; rule-based perturbations show lower quality and consistency win rates (e.g., 55 % vs 28 %)

**Weaknesses:**

1. The “student” LLM ability parameters (θ) are estimated but not analyzed—e.g., how they correlate with model size or reasoning specialization.
2. The difficulty ranker relies on text embeddings; semantic fidelity is measured indirectly. Cases where numerical tweaks superficially raise difficulty but not reasoning depth aren’t deeply analyzed.
3. The paper compares only to rule-based perturbations (e.g., GSM-Plus). It omits baselines like adversarial rewriting via contrastive prompting or reasoning-guided re-sampling (e.g., Math-Perturb 2025).
4. The pairwise ranking and RL training pipeline may be costly for larger datasets (O(N²) pair combinations). Discussion of computational efficiency or approximate ranking would be beneficial.

**Questions:**

1. How do you ensure that rewritten problems truly increase reasoning difficulty rather than only numeric or lexical variation?
2. The total reward combines difficulty, correctness, and keyword/length terms with weights (α, β, γ). How sensitive are results to these weights, and did you observe mode collapse toward trivial or overly verbose rewrites?
3. What fraction of rewrites fail correctness verification without GPT-5-mini supervision? Could open-source verifiers (e.g., DeepSeek-Verifier) achieve comparable filtering?

---

> ### Author Response · Authors · 2025-11-20
> **Author Response to Reviewer zuaw Part 1**
>
> Dear Reviewer zuaw,
>
> We sincerely thank the reviewer for the high-quality feedback and insightful comments. Please allow us to respond to the weaknesses and questions raised.
>
> > **W1: The “student” LLM ability parameters (θ) are estimated but not analyzed—e.g., how they correlate with model size or reasoning specialization.**
>
> We have reported the fitted capability results for 35 subject LLMs in Table 6. Based on these scores, we observe a **positive correlation** between parameter size and capability within the same series (such as Qwen2.5 or Qwen3), although the scaling is not linear. Furthermore, frontier reasoning models consistently demonstrate higher capability. We will include a comprehensive analysis of these trends in the revised version of our paper.
>
>
>
> > **W2: The difficulty ranker relies on text embeddings; semantic fidelity is measured indirectly. Cases where numerical tweaks superficially raise difficulty but not reasoning depth aren’t deeply analyzed.**
>
> Current research has shown that embedding can capture digital information and complex reasoning semantics [1] [2].  Through embeddings, models internally construct structured representations of mathematical concepts, which facilitates their understanding of content ranging from low-level elements to high-level abstractions. Furthermore, our proposed method serves as a general framework. The embedding-based calculation is merely one implementation choice, which can be substituted with other methods for representing mathematical questions to fit IRT difficulties.
>
> [1] Wang P Y, Liu T S, Wang C, et al. A Survey on Large Language Models for Mathematical Reasoning. 2025.
>
> [2] Sivakumar J A, Moosavi N S. How to leverage digit embeddings to represent numbers? 2025.
>
>
>
> Our case study (Figure 3) presents a rewriting example demonstrating that our method is not merely simple numerical and lexical substitution, but is composed of many rewriting strategies that increase reasoning depth, such as adding conditions and changing solution expressions. To systematically analyze these rewrites, we distill 6 most common rewrite patterns from the questions in the RIDE-AIME and RIDE-AMC datasets: (1) Wording and Paraphrasing Modification, (2) Distracting Info, (3) Numerical Substitution, (4) Add Extra Steps, (5) Change Constraints, and (6) Change Target. (Detailed descriptions are provided in Appendix B.4). First, we counted the frequency with which each rewriting strategy appeared in our set of rewritten questions:
>
> | **Type**                                  | **Frequency (%)** |
> | ----------------------------------------- | ----------------- |
> | **Wording and Paraphrasing Modification** | 65.7              |
> | **Insertion of Distracting Information**  | 11.4              |
> | **Numerical Value Substitution**          | 17.1              |
> | **Addition of Extra Variables/Steps**     | 32.9              |
> | **Change of Constraints or Rules**        | 44.3              |
> | **Change of the Target Expression**       | 28.6              |
>
> Secondly, we tallied the distribution of the number of rewriting strategies present in the rewritten questions:
>
> | **Number of strategies** | **Percentage** |
> | ------------------------ | -------------- |
> | 1                        | 35.7           |
> | 2                        | 35.7           |
> | 3                        | 21.4           |
> | 4                        | 7.1            |
>
> Over 60% of the rewritten questions are composed of two or more strategies. To further investigate the impact on reasoning depth, we categorize strategies 1-3 as Surface Strategies, which have a minor impact on reasoning depth, and strategies 4-6 as Deep Strategies, which significantly affect the required reasoning.  We categorize the questions based on the type of strategies used: those employing only Surface Strategies, those employing only Deep Strategies, and those employing a Mixture of both. The frequency statistics are as follows：
>
> | **Type**         | **Frequency (%)** |
> | ---------------- | ----------------- |
> | **Surface-only** | 31.4              |
> | **Mixed**        | 44.3              |
> | **Deep-only**    | 24.3              |
>
> Statistics show that rewrites using a mix of both Surface and Deep strategies are the most common (44.3%).  Therefore, our rewriting data effectively increases intrinsic difficulty (as evidenced by the degradation in model performance).

---

> ### Author Response · Authors · 2025-11-20
> **Author Response to Reviewer zuaw Part 2**
>
> Dear Reviewer zuaw,
>
> We sincerely thank the reviewer for the high-quality feedback and insightful comments. Please allow us to respond to the weaknesses and questions raised. (Part 2)
> > **W3: The paper compares only to rule-based perturbations (e.g., GSM-Plus). It omits baselines like adversarial rewriting via contrastive prompting or reasoning-guided re-sampling (e.g., Math-Perturb 2025).**
>
> Drawing upon the methodology proposed in [1], we designed an experiment to evaluate contrastive prompting. Specifically, we instructed Qwen3-8B to simultaneously generate both an easier and a harder rewrite, selecting the harder version as the final output. The comparative results regarding win rates are presented below:
>
> | **Compared Method**       | **Dataset** | **Average (Baseline)** | **Average (RIDE)** | **Consistent (Baseline)** | **Consistent (RIDE)** |
> | ------------------------- | ----------- | ---------------------- | ------------------ | ------------------------- | --------------------- |
> | **Contrastive Prompting** | AMC-23      | 49.17                  | **50.83**          | 40.00                     | **40.83**             |
> |                           | AIME-24     | 32.78                  | **67.22**          | 26.67                     | **61.11**             |
>
> The results demonstrate that our approach yields higher rewrite quality compared to direct contrastive prompting. Furthermore, regarding dataset construction efficiency, related work such as MATH-Perturb relies on 12 PhD students to manually rewrite large-scale data. This labor-intensive workflow lacks generalizability and is ill-suited for direct transfer to other datasets. In contrast, our method limits human involvement solely to the final verification of the open-source benchmark, with no manual content modification required. The entire pipeline is fully automated by models, ensuring greater efficiency and generalizability.
>
> [1] Liang Yao. Large language models are contrastive reasoners, 2025
>
>
>
> > **W4: The pairwise ranking and RL training pipeline may be costly for larger datasets (O(N²) pair combinations). Discussion of computational efficiency or approximate ranking would be beneficial.**
>
> The computational complexity is **effectively** **optimized by synthesizing pairwise data** within specific sub-categories of mathematical questions. For instance, our 2,000 questions only yielded 209,648 data pairs. Furthermore, as the difficulty ranker employs a lightweight 3-layer MLP, the overall computational overhead remains minimal.
>
>
>
> > **Q1: How do you ensure that rewritten problems truly increase reasoning difficulty rather than only numeric or lexical variation?**
>
> Please refer to our response about W2. Case study (Figure 3a) and statistical result (Figure 3b, line410-415) show that our rewriting data effectively increases intrinsic difficulty (as evidenced by the degradation in model performance) rather than only numeric or lexical variation.
>
>
>
> > **Q2: The total reward combines difficulty, correctness, and keyword/length terms with weights (α, β, γ). How sensitive are results to these weights, and did you observe mode collapse toward trivial or overly verbose rewrites?**
>
> To prevent training collapse, we dynamically adjust the reward weights during the initial phase based on observed training dynamics. Initially, the model tends to generate overly verbose rewrites; however, as the reward signals stabilize, the length reward encourages the model to gradually reduce response length, thereby avoiding redundancy.
>
> Furthermore, we conducted ablation studies to analyze the impact of the difficulty and correctness rewards. First, we evaluate the correctness of the generated rewrites against a baseline model trained **without the correctness signal $R_{cor}$.** Subsequently, we **remove the difficulty reward $R_{diff}$** from the RL objective and compare the resulting model's rewritten questions with our final rewritten questions, again reporting the win rate.
>
> | **Compared Method**       | **Dataset** | **Baseline (Average Win Rate)** | **RIDE (Average Win Rate)** | **Baseline (Consistent Win Rate)** | **RIDE (Consistent Win Rate)** |
> | ------------------------- | ----------- | ------------------------------- | --------------------------- | ---------------------------------- | ------------------------------ |
> | **w/o $R_{\text{diff}}$** | AMC-23      | 43.75                           | **54.17**                   | 35.83                              | **47.50**                      |
> |                           | AIME-24     | 27.78                           | **72.22**                   | 21.11                              | **65.56**                      |
>
> | Model                   | Correctness |
> | :---------------------- | :---------: |
> | RIDE-GPT-5-mini         |  **45.00**  |
> | RIDE w/o $R_\text{cor}$ |    32.50    |

---

> ### Author Response · Authors · 2025-11-20
> **Author Response to Reviewer zuaw Part 3**
>
> Dear Reviewer zuaw,
>
> We sincerely thank the reviewer for the high-quality feedback and insightful comments. Please allow us to respond to the weaknesses and questions raised. (Part 3)
>
>
>
> > **Q3: What fraction of rewrites fail correctness verification without GPT-5-mini supervision? Could open-source verifiers (e.g., DeepSeek-Verifier) achieve comparable filtering?**
>
> Please refer to our response about Q2. Besides, we employ the **open-source model DeepSeek-V3.2** as a verifier to provide correctness rewards for training **RIDE-DeepSeek**. To assess the impact of this reward, we evaluate the correctness of the generated rewrites against a baseline model trained without the correctness signal:
>
> | Model                   | Correctness |
> | :---------------------- | :---------: |
> | RIDE-GPT-5-mini         |  **45.00**  |
> | RIDE-DeepSeek           |  **37.50**  |
> | RIDE w/o $R_\text{cor}$ |    32.50    |
>
> Results show that while DeepSeek-V3.2 yields lower correctness than GPT-5-mini, it still offers a valid reward signal, resulting in **higher correctness** than the baseline without any reward.
>
> To test the filter,  we conducted an experiment to evaluate whether using the filter model GPT-5-mini and DeepSeek-V3.2, could accurately judge the correctness given a mathematical question(without standard answers), its generated solution, and its final answer. We provided 2,000 math questions , along with the solutions and answers generated by two test models: Qwen2.5-Math-7B and Gemma3-12B. The results are as follows:
>
> | Response Source     | Verifier      | Precision | Recall | F$_1$ | Accuracy |
> | :------------------ | :------------ | :-------: | :----: | :---: | :------: |
> | **Qwen2.5-Math-7B** | GPT-5-mini    |   97.96   | 79.17  | 87.57 |  85.70   |
> |                     | Deepseek-V3.2 |   89.48   | 62.19  | 73.38 |  71.30   |
> | **Gemma3-12B**      | GPT-5-mini    |   98.38   | 79.71  | 88.06 |  86.00   |
> |                     | Deepseek-V3.2 |   93.84   | 54.09  | 68.62 |  67.95   |
>
> DeepSeek verifier achieved precision of nearly **90%**. This indicates that  DeepSeek can also precisely assign low reward signals to incorrect solutions and are effective for filtering erroneous data. The relatively low recall, however, implies that the verifiers tend to be conservative, flagging an otherwise correct solution and answer as incorrect.  Thus, even with the presence of false negatives, DeepSeek-V3.2 successfully filters out incorrect data, making it a feasible choice for a filter model.
>
> We have marked the added content in **blue** in the revised paper. Thank you again for your careful reading and valuable review.

---

> ### Author Response · Authors · 2025-11-26
>
> Dear Reviewer zuaw,
>
> I hope this message finds you well. As the discussion period is nearing its end with **less than one week,** I wanted to ensure that we have addressed all your concerns satisfactorily. If there are any additional points or feedback you'd like us to consider, please let us know. Your insights are invaluable to us, and we're eager to address any remaining issues to improve our work.
>
> Thank you for your time and effort in reviewing our paper.

---

### Author Response · Authors · 2025-12-01
**Summary of Rebuttal Phase & Reviewer Consensus**

Dear Program Chairs, Senior Area Chairs, and Area Chairs,

We express our sincere gratitude to all reviewers for their insightful feedback and the time dedicated to reviewing our work. We provide this summary of the rebuttal phase to highlight the consensus on our contributions and the substantial improvements made to the paper based on their suggestions.

## **1. Consensus on Strengths**

We are highly encouraged that the reviewers recognized the novelty, rigor, and significance of our RIDE framework:

- **Innovative Methodology:** Reviewers **zuaw** and **be6K** commended the novel integration of **Item Response Theory (IRT)** for formalizing and quantifying "question difficulty," describing the methodology as "rigorous", "well-motivated", and "well-conceived." Reviewer **iD67** also noted that the IRT model handsomely credits the hardness of generated questions.
- **Extensive Evaluation:** Reviewers **zuaw** and **be6K** highlighted the comprehensive experimental coverage across **23-35 LLMs** (spanning various families like Qwen, Llama, GPT, etc.) as a major strength, providing strong evidence for the brittleness of current models.
- **Clarity & Pipeline:** Reviewers **GWMD** and **iD67** recognized the clarity of the **RIDE pipeline** and the clear presentation of the workflow. Reviewer **GWMD** specifically praised the originality of the rewriting prompt and the strategy of using VAE/Sampling augmentation to mitigate data scarcity.

## **2. Revisions & Responses**

We have conducted extensive additional experiments and analyses to address the raised concerns comprehensively:

**Robustness of IRT & Difficulty Estimation**

- **Stability Analysis (Reviewer be6K, iD67):** To address concerns about potential bias from the specific student pool, we added experiments incorporating weaker models (e.g., Qwen3-1.7B) and additional augmentation. Results showed high stability (Spearman correlation > 0.99) and confirmed that our IRT difficulty aligns with human-rated difficulty to a similar extent as empirical error rates, thereby validating its reliability.

- **Taxonomy & Distribution (Reviewer zuaw, be6K, GWMD):** We distilled 6 distinct rewriting patterns (e.g., Distracting Info, Change Constraints) and categorized them into "Surface" vs. "Deep" strategies. Statistics show a balanced mix (44.3% mixed strategies).
- **Decoupling Difficulty from Robustness (Reviewer GWMD):** We conducted **McNemar’s test** and difficulty-stratified analysis on "Surface-only" rewrites. The significant p-value ($2.38 \times 10^{-11}$) confirms that performance drops are primarily due to model **non-robustness.**

**Effectiveness of RL & Verifiers**

- **RL Ablation Studies (Reviewer zuaw, GWMD):** We added comprehensive ablation studies (comparing RIDE vs. SFT-only, and RIDE w/o $R_{diff}$/$R_{cor}$). Results in the **revised Table 2 and Table 5** demonstrate that the RL stage significantly enhances both the difficulty and correctness of the rewrites.
- **Verifier Precision (Reviewer zuaw, be6K):** We evaluated the precision of our filter models (GPT-5-mini and DeepSeek-V3.2), achieving 98% precision. This confirms that our automated pipeline effectively filters incorrect data, **minimizing the need for human intervention.**

**Baselines & Comparisons**

- **Comparison with Baselines (Reviewer zuaw):** We added a direct comparison with **Contrastive Prompting**, where RIDE demonstrated superior rewrite quality and win rates.

We would like to kindly bring to your attention that we have carefully answered the questions from Reviewers, but has **not yet received all responses** from them during the rebuttal period. Given the importance of their feedback in the final decision-making process, we were wondering if it might be possible to **consider the situation accordingly.**

We fully understand that reviewers are managing busy schedules, and we truly appreciate all the efforts you and the reviewers have already dedicated to the review process.

---

### Meta-Review · Area_Chair_fY3x · 2026-01-17

**Summary:**

The paper proposes to finetune a model to rewrite existing math reasoning questions into more difficult perturbations of the original questions. The proposed model, "RIDE", is trained to maximize a reward composed of (1) a difficulty reward defined by a difficulty ranker trained using difficulty estimated by Item Response Theory (IRT) model, (2) a GPT5-mini-as-a-judge defined reward of question's correctness, and (3) a keyword reward and length penalty. Evaluation of 35 LLMs shows a significant performance drop on 26 LLMs. This reveals the non-robustness of the models in the difficulty-evolved benchmark of questions.

**Reviewer Concerns:**

- Unclear whether it is possible and how to decouple the difficulty evolution and robustness as two confounders of the performance drop.
- The method heavily relies on GPT5-mini to define the RL reward and to evaluate the quality of the rewritten questions. However, it is not clear how reliable GPT5-mini performs as a reward function/judge, when compared to other LLMs. Moreover, using GPT5-mini to provide both training rewards and filtering/evaluation in test leads to a circular dependency, which can be problematic.
- Lack of systematic analysis of the patterns of rewriting and whether they truly increase the reasoning difficulty.
- The LLM difficulty captured by IRT might be divergent from human-perceived difficulty.
- The VAE-based sampling for augmentation of response matrix rows in IRT might be harmful.
- The paper only compares their model with rule-based perturbation but ignores others, such as contrastive prompting.
- Potentially expensive cost of pairwise difficulty ranker.
- Lack of an ablation study and sensitivity analysis of the three weighted-averaged rewards.
- While the paper claims an automatic difficulty evolving approach, they still rely on human annotators to verify the quality and correctness of the questions in the final stage (before being used for benchmarking purposes).
- Lack of an ablation study for the contribution of RL and IRT reward.

**Reviewer Scores:**

- The initial and the final ratings from the reviewers are 4, 4, 6, 6, with confidence 3, 4, 3, 4, making the paper a boarderline.
- Only one reviewer with a rating 4 (confidence 3) participated in the discussion and raised follow-up questions, but the authors' further response did not make the reviewer raise the final rating.
- I carefully checked all the questions and responses from the authors. The authors provided several new experimental results to address the concerns, with detailed clarifications and analysis. Most of them are successful; for example, the systematic analysis of the rewriting patterns is thorough and insightful.
- However, the top-2 concerns above have not been fully addressed by the response. In particular, the performance drop can still be a result of both non-robustness and increased difficulty, and it is still challenging to separate their effects.
- Moreover, the circular dependency of GPT5-mini in both training and evaluation can make several main claims of advantages in this paper ineffective, because they heavily rely on the win rate between two questions, and the judge is always GPT5-mini. I feel it is risky to accept the paper before excluding the possibility. In other words, it could be a biased evaluation as the training aims to maximize the reward from the judge in the evaluation.
- Therefore, I lean towards rejecting this paper and hope the authors can address these issues and submit it to the next conference.

---

### Decision · Program_Chairs · 2026-01-26

Reject